# Unraveling the dynamics of conductive filaments in MoS$_2$-based memristors by *operando* transmission electron microscopy

Ke Ran [1,2,3] ✉, Janghyun Jo[3], Sofía Cruces[4], Zhenxing Wang [1], Rafal E. Dunin-Borkowski [3], Joachim Mayer [2,3] & Max C. Lemme [1,4] ✉

Advanced *operando* transmission electron microscopy (TEM) techniques enable the observation of nanoscale phenomena in electronic devices during operation. Here, we investigated lateral memristive devices composed of two dimensional layered MoS$_2$ with Pd and Ag electrodes. Under external bias voltage, we visualized the formation and migration of Ag conductive filaments (CFs) between the two electrodes, and their complete dissolution upon reversing the biasing polarity. The CFs exhibited a wide range of sizes, from several Ångströms to tens of nanometers, and followed diverse pathways: along the MoS$_2$ surfaces, within the van der Waals gap between MoS$_2$ layers, and through the spacing between MoS$_2$ bundles. Our method enables correlation between current-voltage responses and real-time TEM imaging, offering insights into failed and anomalous switching behaviors, and clarifying the cycle-to-cycle variabilities. Our findings provide solid evidence for the electrochemical metallization mechanism, elucidate the formation dynamics of CFs, and reveal key parameters influencing the switching performance.

Neuromorphic computing (NC) emulates the efficiency, versatility, and resilience of the human brain in terms of massive parallel learning and computing[1–3], and is expected to enable real-time cognitive tasks[4]. Among the technologies explored for next-generation NC chips, memristive devices have emerged as leading candidates, offering low energy consumption, high-density integration, rapid switching speeds, high endurance, and compatibility with complementary metal-oxide-semiconductor (CMOS) fabrication processes[5–7]. Such "memristors" are typically two-terminal devices with two electrodes connected through a switching medium. A particularly promising class of memristors is the electrochemical metallization (ECM) type[8,9], which uses oxide bulk materials[10–12] or two-dimensional (2D) materials[13–17] as the switching medium. These devices operate through the formation and dissolution of conductive filaments (CFs) within the switching medium under an applied electric field, thereby enabling reversible resistive switching, and have

demonstrated exceptional potential[10,18–20]. Their switching happens during a "SET" process, where a bias voltage applied to the electrodes, induces the formation of CFs that result in a low resistance state (LRS) of the device. Conversely, a "RESET" process can be induced by reversing the bias voltage polarity, leading to the dissolution of the CFs and restoring the device to a high resistance state (HRS). The key performance metrics for memristors include the switching voltage, ON/OFF ratio, switching time, and switching energy[21].

2D materials such as molybdenum disulfide (MoS$_2$) have attracted considerable attention as the switching material of ECM memristors. Their layered structure, atomic thickness, high surface area, and unique electronic properties[22] make them promising candidates for achieving reliable switching performance and scalability in practical NC applications[23,24]. However, memristors based on 2D materials often exhibit nonuniform and inconsistent switching behaviors[25–29], which is attributed primarily to unpredictable ion transport. Therefore, a

[1]Advanced Microelectronic Center Aachen, AMO GmbH, Aachen, Germany. [2]Central Facility for Electron Microscopy GFE, RWTH Aachen University, Aachen, Germany. [3]Ernst Ruska-Centre for Microscopy and Spectroscopy with Electrons ER-C, Forschungszentrum Jülich GmbH, Jülich, Germany. [4]Chair of Electronic Devices ELD, RWTH Aachen University, Aachen, Germany. ✉e-mail: ran@amo.de; max.lemme@rwth-aachen.de

comprehensive understanding of the dynamics governing the formation and dissolution of CFs within 2D materials is essential for optimizing the performance and reliability of 2D material-based ECM memristors.

Various experimental techniques, including conductive atomic force microscopy (CAFM)[13,30], Raman spectroscopy[31], scanning tunneling microscopy (STM)[31], and transmission electron microscopy (TEM)[32,33,] have been employed to investigate CF dynamics. However, these ex situ methods typically compare only the HRS and LRS, lacking the ability to provide real-time observations of the processes occurring during the SET and RESET operations. In contrast, *operando* TEM, with its high spatial and temporal resolution, coupled with versatile sample manipulation capabilities, offers a powerful approach for real-time visualization of CF dynamics under external bias[34–36]. While several studies have reported the nucleation and growth of metal clusters[8,37–39], these efforts have focused predominantly on oxide-based memristors[10,11,40], with limited exploration of 2D material-based devices[20,41]. Consequently, real-time observations that correlate the electric device performance with the CF dynamics within layered 2D materials are lacking.

In this article, we present the *operando* TEM investigation of resistive switching in a lateral memristive device with the 2D MoS$_2$ as the switching material. Upon successful switching, CFs of varying sizes were observed along the MoS$_2$ surface, within van der Waals (vdW) gaps, between MoS$_2$ bundles, and across disrupted MoS$_2$ layers. The formation and dissolution of these CFs were directly correlated with the device behavior through simultaneous TEM imaging and current-voltage (*I-V*) measurements. Our approach offers unique insights into cycle-to-cycle variabilities and facilitates comparisons across different devices. Our findings elucidate the microscopic origin of resistive switching and provide general guidance for designing novel memristive devices.

## Results and discussion
### Device for *operando* experiments
Lateral MoS$_2$ memristive devices with palladium (Pd) and silver (Ag) electrodes were used in this work[42], as shown in the schematic in Fig. 1a. The devices were fabricated from multilayer MoS$_2$ grown via metal-organic chemical vapor phase deposition (MOCVD) on 2″ sapphire wafers[43]. The MoS$_2$ was then wet transferred onto a Si/SiO$_2$ substrate[44,45]. After photolithography, Pd and Ag electrodes with a thickness of approximately 50 nm were fabricated on top of the MoS$_2$ via electron beam evaporation and a standard lift-off process. Since Ag is rather chemically reactive (Supplementary Note 1 and Supplementary Fig. 1), an extra aluminum (Al) layer of ~50 nm was evaporated on top of the Ag electrode. The lateral distance between the electrodes was approximately 1–2 μm, making up the MoS$_2$ channel of the device (see the thick quadrilateral in Fig. 1a). The fabrication was finished by etching the excess MoS$_2$ layers outside the channel region via reactive ion etching. An insulating aluminum oxide (Al$_2$O$_3$) layer with a thickness of 80 nm was deposited by electron beam evaporation onto the channel region to facilitate the subsequent focused ion beam (FIB) preparation and *operando* TEM. Figure 1b shows a scanning electron microscopy (SEM) image of the device from the top, where the deposited Al$_2$O$_3$ is outlined by the dashed rectangle. Details of the fabrication process can be found in the "Methods" section.

FIB lamellas comprising the metal electrodes and the MoS$_2$ channel were prepared, as outlined by the solid rectangle in Fig. 1b. A schematic illustration of the *operando* TEM setup is shown in Fig. 1c, d[34]. A grounded and mobile gold (Au) tip was used to contact the FIB lamella, which was attached to a fixed TEM grid where external bias voltage was applied. A closeup of the tip/lamella contact is further illustrated in Fig. 1d. The high-angle annular dark-field (HAADF) image in Fig. 1e shows one lamella (L1) with the Au tip inside the TEM. The channel region in Fig. 1e is further enlarged in Fig. 1f, where the Pd and

Ag electrodes are visible on both sides. The channel between them is estimated to be ~800 nm, and the lamella thickness is less than 100 nm in the y direction (Supplementary Note 2 and Supplementary Fig. 2). The corresponding composite map of Pd, S, and Ag is shown in Fig. 1g. It was obtained by energy-dispersive X-ray spectroscopy (EDXS) elemental mapping before any electrical measurements (Supplementary Note 3 and Supplementary Fig. 3) and shows a good agreement with the intended device design. Close to the Ag electrode, a few isolated Ag particles can be detected, possibly resulting from the FIB preparation. This setup allows simultaneous biasing and imaging and thus enables real-time observation of the CF dynamics during switching. In our devices, both multilayer MoS$_2$ (Fig. 1h) and MoS$_2$ bundles (several stacks of multilayer MoS$_2$ as in Fig. 1i and from L1) are present. In Fig. 1h, the spacing between neighboring MoS$_2$ layers was consistently measured as ~6.4 Å, in line with reports in the literature[46,47], whereas the thickness of MoS$_2$ decreases by ~1.5 nm from left to right. The three MoS$_2$ bundles in Fig. 1i are separated by several nms, and each bundle contains approximately 5–10 layers.

### Bipolar resistive switching
The electrical properties of L1 in Fig. 1e–g were first checked by applying a small bias voltage sweep of 0 V → 0.1 V → 0 V → −0.1 V → 0 V, with parameters of 0.03 V/step, 0.03 s/step, and a current compliance (CC) of $10^{-4}$ A. The *I-V* characteristics in Fig. 2a show no resistive switching. The HAADF image and the composite elemental map of Pd, S, and Ag in Fig. 2d were recorded immediately after the sweep cycle in Fig. 2a, and they match those in Fig. 1f, g and Supplementary Fig. 3. The enlarged map at the bottom of Fig. 2d further confirms that no significant amount of Ag can be detected within the channel. Next, a bias voltage sweep with a larger range of 0 V → 5 V → 0 V → −5 V → 0 V was applied with 0.05 V/step, 0.03 s/step, and CC = $10^{-4}$ A. The *I-V* characteristics in Fig. 2b show that the lamella was switched on by the 0 V → 5 V sweep, remained at the LRS under the 5 V → 0 V sweep, was switched off by the 0 V → −5 V sweep, and remained at the HRS afterwards. The measured resistance change is approximately one order of magnitude. Again, HAADF imaging and EDXS elemental mapping were applied directly after the sweep cycle, as shown in Fig. 2e and Supplementary Fig. 4. Except for an evidently reduced Ag electrode, the difference between Fig. 2d, e is rather negligible. Figure 2c shows the *I-V* characteristics of the lamella in response to a 0 V → 5 V → 0 V sweep, which switched the device on and left it in the LRS. The corresponding HAADF image in Fig. 2f shows obvious differences from Fig. 2e: the MoS$_2$ bundle at the bottom appears much brighter in Fig. 2f, both in the middle of the channel and close to the electrodes. In addition, the size of the Ag electrode is reduced. The composite elemental map in Fig. 2f confirms significant Ag signals, thus Ag filaments, between the MoS$_2$ bundles (Supplementary Note 4 and Supplementary Fig. 5). Along the two dotted lines in Fig. 2e, f, both the HAADF and Ag intensity profiles were extracted, and plotted in Fig. 2g in black and green, respectively. There are two peaks along the HAADF profile taken from the image in Fig. 2e (marked by the pair of gray lines in Fig. 2g), corresponding to the two MoS$_2$ bundles (see the top left inset in Fig. 2e where the middle bundle is locally broken), whereas no significant Ag peaks can be observed. In contrast, the HAADF profile from Fig. 2f shows an additional strong peak (denoted by the arrow in Fig. 2g) between the two MoS$_2$ bundles (the top left inset in Fig. 2f). Coincidently, a significant Ag peak is detected as well. Thus, the increased HAADF contrast can be directly associated with Ag filament formation in our system. Since EDXS elemental mapping is usually time-consuming (up to one hour to collect sufficient counts), while resistive switching is a relatively fast process (several seconds), HAADF imaging with sub-second frame time can thus efficiently monitor the Ag filament during switching and was utilized in the following analysis. In Fig. 2h, we further investigated the upper right regions in the HAADF images as marked by the dashed rectangles in Fig. 2d–f. In particular,

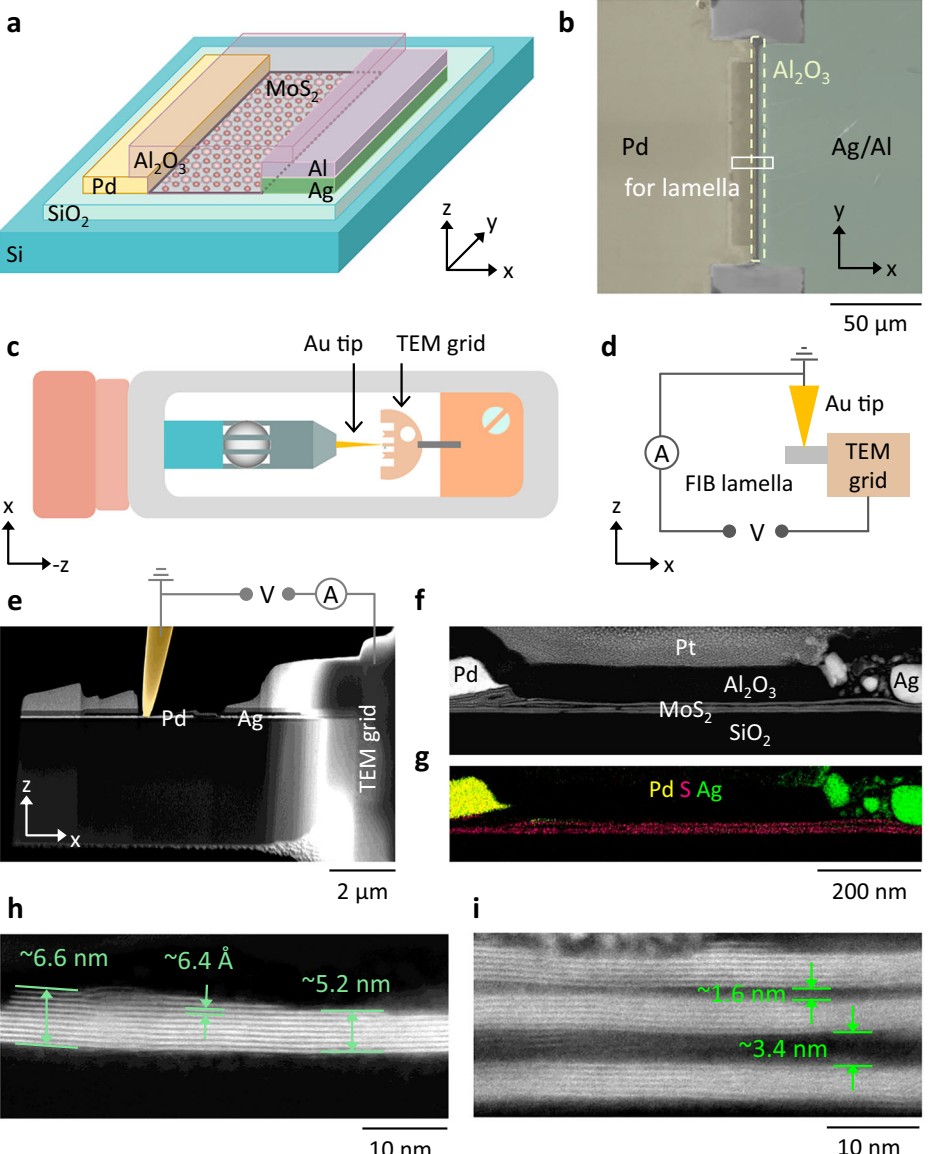

**Fig. 1 | Memristive device for *operando* transmission electron microscopy (TEM). a** Design of the lateral memristor based on MoS$_2$. The MoS$_2$ was first transferred to the Si/SiO$_2$ substrate, followed by electrode deposition (Pd, Ag, and Al, ~50 nm thick). The MoS$_2$ channel between Pd and Ag/Al (outlined by the thick quadrilateral) is approximately 1–2 μm in the x direction. Additionally, ~80 nm thick Al$_2$O$_3$ was deposited onto the channel. **b** Top-view scanning electron microscopy (SEM) image of the lateral device. **c, d** Schematic of the TEM holder and a closeup around the Au tip/FIB lamella contact. The Au tip is grounded and mobile, while the lamella is attached to the fixed TEM grid, through which external biasing can be applied. **e** High-angle annular dark-field (HAADF) image showing the lamella and the *operando* setup inside the TEM. **f, g** HAADF image and the corresponding composite map of Pd, S, and Ag based on energy-dispersive X-ray spectroscopy (EDXS) elemental mapping showing the channel region in (**e**). **h, i** HAADF images showing the cross-section of transferred MoS$_2$.

we outlined and compared the shapes of both a (random) isolated Ag particle and the Ag electrode. While the shape of the Ag particle is relatively stable among the three images, the size of the Ag electrode continuously decreases, confirming the movement of Ag ions from the electrode into the device channel.

**Dynamics of the Ag filaments**

Taking advantage of the *operando* TEM setup, the dynamics of the Ag filaments during a biasing sweep can be monitored by recording a series of HAADF images simultaneously. For the case of a complete switching cycle, as shown in Fig. 2b, the bias voltage lasted for 12 s, while 95 images were recorded with a frame time of ~0.13 s. Selected images taken at different times are shown in Fig. 3a (the image contrast was set to saturation to emphasize Ag within the channel, see Supplementary Fig. 6). Qualitatively, the MoS$_2$ bundles in the channel are already brighter at ~2.3 s (as noted by the triangles) than at ~1.7 s. The increased contrast is visible until ~7.1 s and starts to decrease at ~7.9 s. The image at ~8.5 s is almost comparable with that at ~1.7 s, and it remains that way until the biasing sweep ended. Moreover, the shape of the Ag electrode varied accordingly, i.e., it shrank continuously until 7.1 s and then expanded again (Supplementary Video 1). To analyze the switching process more quantitatively, four areas (A1–A4) were defined at the bottom in Fig. 3a, as outlined by the dotted rectangles. For each HAADF image in the series, the intensities within the four areas were integrated ($A$) and compared with respect to the average values obtained from the initial 6 images ($\bar{A}$). We plotted the intensity variations $\Delta A$ (defined as $\Delta A = \frac{A - \bar{A}}{\bar{A}}$) as a function of time for A1–A4 in Fig. 3b. The corresponding bias voltage is indicated at the top axis. $\Delta A1$ of the Pd electrode and $\Delta A3$ of the isolated Ag particle were rather stable during the biasing sweep (Fig. 3b, bottom). In contrast, a

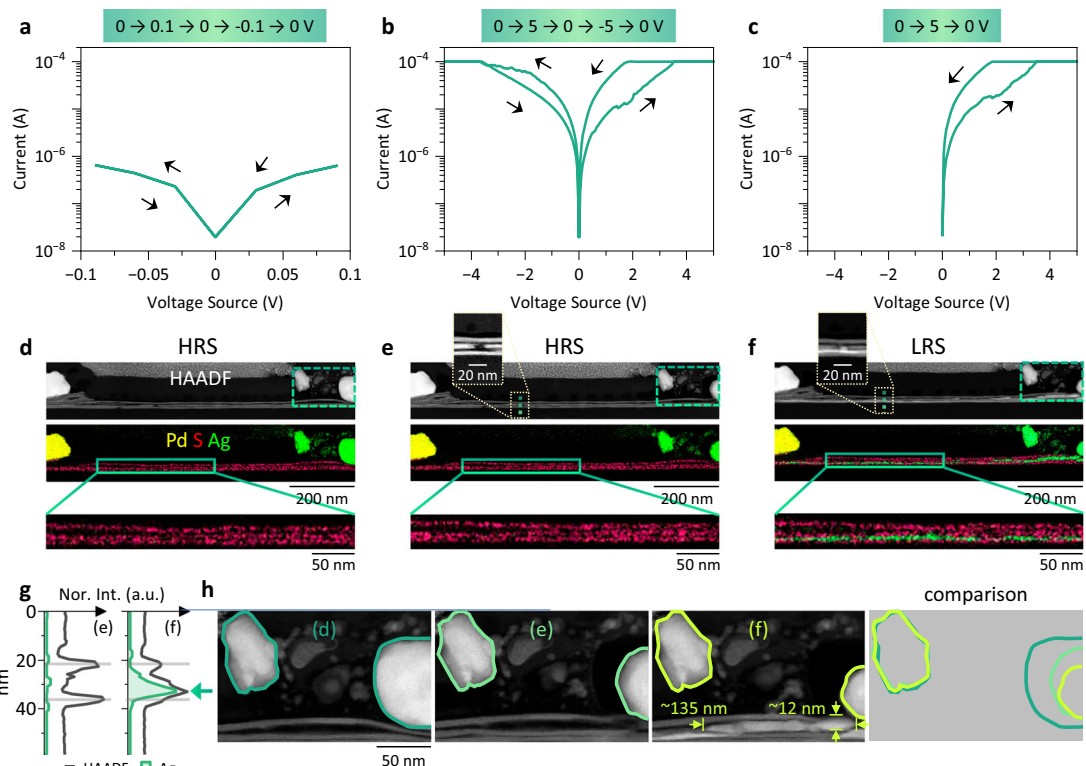

**Fig. 2 | Lamella under various biasing conditions. a–c** The *I-V* responses of the same lamella (L1) upon three bias voltage sweeps, as indicated at the top. **d–f** HAADF images and the corresponding composite maps of Pd, S, and Ag based on EDXS elemental mapping acquired directly after (**a–c**), where L1 was left at high resistance state (HRS) or low resistance state (LRS). At the bottom are enlarged maps from the regions outlined by the solid rectangles. The upper left insets in

(**e** and **f**) are enlarged images from the regions outlined by the dotted rectangles with adjusted contrast limit. **g** HAADF and Ag intensity profiles along the two dotted lines in (**e** and **f**). Two pairs of gray lines mark the same positions. **h** Enlarged images from the dashed rectangles in (**d–f**). Both the isolated Ag particle (left) and the Ag electrode (right) are outlined, and their shapes are compared. A thick Ag filament (-12 × 135 nm) fills the space between the two MoS₂ bundles.

trapezoidal shape is observed for ΔA2 in the MoS$_2$ channel, whereas ΔA4 of the Ag electrode shows a similar but inverted curve. An abrupt jump of ΔA2 took place between 2.75 V and 3.8 V within -0.6 s. Afterwards, ΔA2 remained at a high value for -4.8 s until a negative bias voltage of -−1.85 V was reached. Stimulated by the negative biasing sweep between −1.85 and −4.18 V, ΔA2 decreased back to its original value within -1.4 s. The corresponding images are shown in Fig. 3a. Here, the RESET process took 1.4 s, whereas the SET process took only 0.6 s. The evolution of ΔA4 of the Ag electrode shows an opposite pattern to that of ΔA2, which proves that it serves as a reservoir for the Ag filaments during switching. In contrast to ΔA2, which returned to its initial value after the complete switching cycle, ΔA4 was reduced by -14%. It also showed a small jump of -2% and a drop of -5% before and after switching, as indicated by the shadowed areas in Fig. 3b.

**Local structural changes of MoS$_2$**
In addition to monitoring the switching with a large field of view, as shown in Fig. 3, the local structural changes in MoS$_2$ caused by biasing sweep were investigated as well. We prepared another lamella (L2), where a single bundle/multilayer MoS$_2$ containing 5–10 layers was identified (Supplementary Note 5 and Supplementary Fig. 7). We successfully induced nonvolatile switching with bias voltage sweeps of 0 V → 10 V → 0 V → −10 V → 0 V with 0.1 V/step, 0.05 s/step, and a CC = 6 × 10$^{-5}$ A. Each sweep lasted for 20 s, while a series of 25 HAADF images with a longer frame time of -0.8 s was recorded to resolve the local structures.

Figure 4a shows the first image from such an image series, where 7 layers of MoS$_2$ were locally monitored (Supplementary Note 6, Supplementary Fig. 8, and Supplementary Video 2). The hypothesis for this local region is that Ag filaments traverse mainly through the vdW gaps,

i.e., the distance *D* between adjacent MoS$_2$ layers, as labeled in Fig. 4a. However, owing to the limited size of *D*, the amount of Ag filaments is rather low, which makes it hard to observe either by HAADF imaging or EDXS elemental mapping. Alternatively, the intercalation of Ag filaments into the vdW gap is expected to cause a certain expansion of *D*, and the fast Fourier transform (FFT) of the recorded HAADF image is quite sensitive to small changes in D. Thus, estimating the mean value of *D*, $D_m$, from each FFT offers an indirect approach to monitor the local evolution of the Ag ions (and their filaments). Figure 4b shows the FFT calculated from Fig. 4a. The spot corresponding to $D_m$ is outlined by a red square, where $d_r = 1/D_m$ ($d_r$ is the distance to the central spot, as defined in Fig. 4b). Figure 4c shows the $d_r$ spot recorded at different times during the switching. Obviously, it is moving. Figure 4d then plots the relative change in each $D_m$, $\Delta D_m$ ($\Delta D_m = \frac{D_m - \bar{D}_m}{\bar{D}_m}$, where $\bar{D}_m$ is the $D_m$ averaged from the first four images), as a function of time. The corresponding bias voltage is indicated on the top axis. Overall, the $D_m$ first increased to 10% and then slowly recovered. As indicated by the two gray lines in Fig. 4d, $D_m$ increased between 3.2 s and 14.4 s, corresponding to the biasing sweep from 6.4 V to −8.8 V, where the device should be in the LRS. Thus, the increased $D_m$ can be associated with the Ag filaments formed along the vdW gaps in the LRS, which is also consistent with the image intensity variation (Supplementary Note 7 and Supplementary Fig. 8). Once the device returned to the HRS upon application of a negative bias voltage, $\Delta D_m$ was marginally above zero. We attribute this to a small amount of residual Ag ions in the vdW gaps or a slow recovery of the MoS$_2$ structure, both of which will contribute to deteriorated device performance. In addition, the deviation of $\Delta D_m$ is relatively large as indicated by the shadow in Fig. 4d, especially at the LRS and the following HRS. This suggests that the formation and

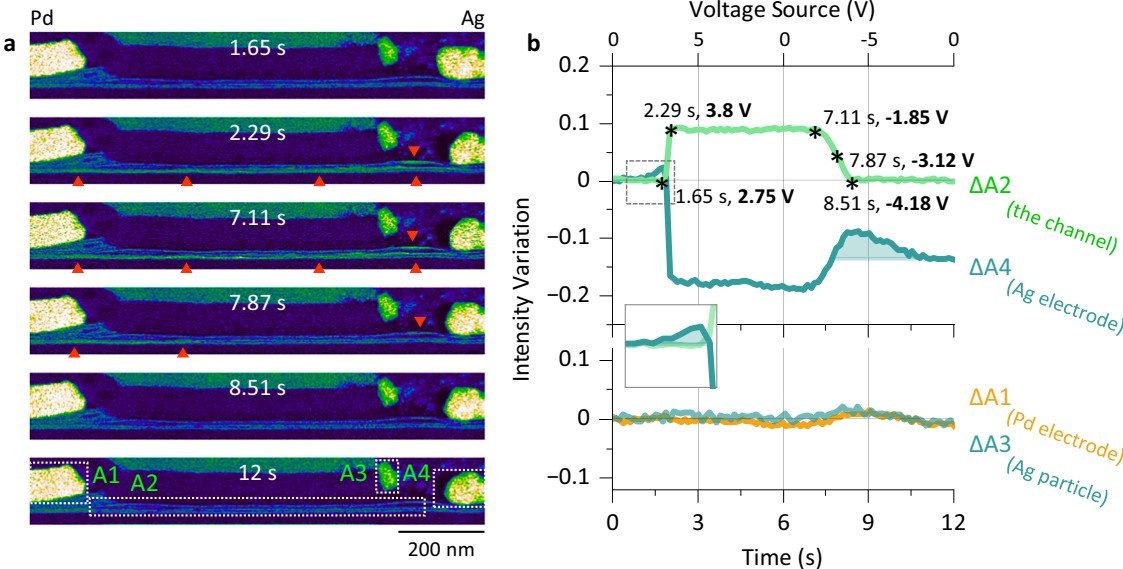

**Fig. 3 | Monitoring the channel during a biasing sweep. a** False color HAADF images acquired at different times during the $0\,V \rightarrow 5\,V \rightarrow 0\,V \rightarrow -5\,V \rightarrow 0\,V$ sweep in Fig. 2b. Four regions (A1–A4) are defined at the bottom, corresponding to the Pd electrode, the $MoS_2$ channel, the isolated Ag particle, and the Ag electrode, respectively. The red triangles mark the emerging contrast from Ag filaments. **b** Intensity variation within the four defined areas (ΔA1–ΔA4) as a function of time during the sweep. The corresponding bias voltage is indicated on the top axis. The inset is an enlarged image from the region defined by the dashed rectangle.

disruption of Ag filaments are rather local and dynamic processes. Combining all the experimental observations, Fig. 4e proposes a possible scenario for the structural evolution of $MoS_2$ during switching. Starting from a nearly perfect structure, the $MoS_2$ layers are largely deformed due to the migration of Ag ions and the resulting filaments, leading to an increased $D_m$ with considerable deviations. After successful RESET, most of the Ag filaments are dissolved, leaving the $MoS_2$ with some residual Ag ions and minor structural modifications from its previously near-parallel layered structure.

The expanded $D$ was also imaged directly with a much longer frame time of ~25 s. Figure 4f shows the HAADF image from L1 recorded after the $0\,V \rightarrow 5\,V \rightarrow 0\,V$ sweep in Fig. 2c, where the lamella remained in the LRS. Three bundles of $MoS_2$ can be distinguished, #1 to #3. The three arrows A, B, and C in Fig. 4f indicate the direction of the image intensity profiles plotted in Fig. 4g. For most of the regions, $D$ is constantly measured as ~6.4 Å. Along A, a gap of ~26.7 Å is estimated, resulting from the local spacing between bundles #1 and #2. The bright contrast within bundle #3 suggests significant formation of Ag CFs after SET. These formed Ag CFs may intercalate into the vdW gaps (Supplementary Note 7 and Supplementary Fig. 9) and lie on the surface of the $MoS_2$ (parallel to the x-z plane as defined in Fig. 1e). The resulted contrast then leads to the strong and broad peaks along B and C, which are measured as ~27.6 Å (4 × 6.90 Å) and ~28.5 Å (4 × 7.13 Å), respectively. Thus, up to 11% expansion of the vdW gap can be estimated due to the intercalation of Ag into adjacent $MoS_2$ layers, in line with the estimation in Fig. 4d.

The $MoS_2$ in our prepared lamellas was often observed to be locally broken, possibly due to the FIB fabrication, the wet transfer, or the defective growth. During a $0\,V \rightarrow 10\,V \rightarrow 0\,V \rightarrow -10\,V \rightarrow 0\,V$ bias voltage sweep applied to L2 (Supplementary Fig. 10 and Supplementary Video 3), a disrupted part of the $MoS_2$ layers was monitored, and selected HAADF images are shown in Fig. 4h. Initially, there was a gap of ~7.2 nm without $MoS_2$. At 3.2 s, the gap became much brighter, suggesting the accumulation of Ag. In addition, as indicated by the triangles, strong intensities were also observed at the top and bottom surfaces of the $MoS_2$ layers. As the switching continued, the maximum intensity around the gap was observed at 9.6 s, where Ag can also be located between neighboring $MoS_2$ layers and thus the

vdW gap (the triangle on the right). At 13.6 s with negative bias voltage, the Ag contrast fades away, and at 14.4 s, the image looks rather similar to that recorded at the beginning. Therefore, in addition to traveling along the surface and vdW gaps of $MoS_2$, Ag filaments can also bridge disrupted $MoS_2$ layers and contribute to resistive switching (see also Fig. 2g). The Ag filaments bridging the disrupted $MoS_2$ are noticed much thicker and have more random shapes than the those along the vdW gaps.

## Cycle-to-cycle variability

To gain insights into the origins of cycle-to-cycle variability, four consecutive $0\,V \rightarrow 5\,V \rightarrow 0\,V \rightarrow -5\,V \rightarrow 0\,V$ bias voltage sweeps were applied to L1 with 0.05 V/step, 0.03 s/step, and a CC $= 10^{-4}$ A. Simultaneously, a series of HAADF images with a frame time of 0.13 s was recorded (Supplementary Video 4).

Figure 5a–l compare the *I-V* responses together with intensity variations within A2 and A4, as defined in Fig. 5m, among the four cycles. During cycle #1, the lamella went through a successful SET process but failed at the RESET with negative biasing. This failure is also reflected in Fig. 5b, c, where ΔA2 does not return to zero but retains a final increase of ~6% (the maximum increase during SET is ~9%), and ΔA4 decreases by ~19%. Thus, most of the Ag filaments formed during SET remain in the channel instead of returning to the Ag electrode (Supplementary Fig. 11). In addition, the inset in Fig. 5a, enlarged from the region outlined by the dashed rectangle, shows an abrupt current jump at approximately −2.3 V (marked by the asterisk). Moreover, there is a bump along ΔA2 at approximately −2.3 V, indicated by the shadowed region in Fig. 5b and enlarged in the inset. This suggests that more Ag filaments can form unexpectedly (since A2 corresponds to the intensity integrated within the whole channel) even during the RESET process and cause abrupt current jumps in the *I-V* curve.

For cycle #2, ΔA2 in Fig. 5e starts with a value of ~0.06 because of the residual Ag within the channel. Similar to Fig. 5b, a sudden jump in ΔA2 occurs at ~2.75 V. The maximum ΔA2 in Fig. 5e is also greater than in Fig. 5b, suggesting that thicker Ag filaments formed during cycle #2. An unexpected current drop can be observed at ~2.95 V in the inset of Fig. 5d, which shows an enlarged plot from the region

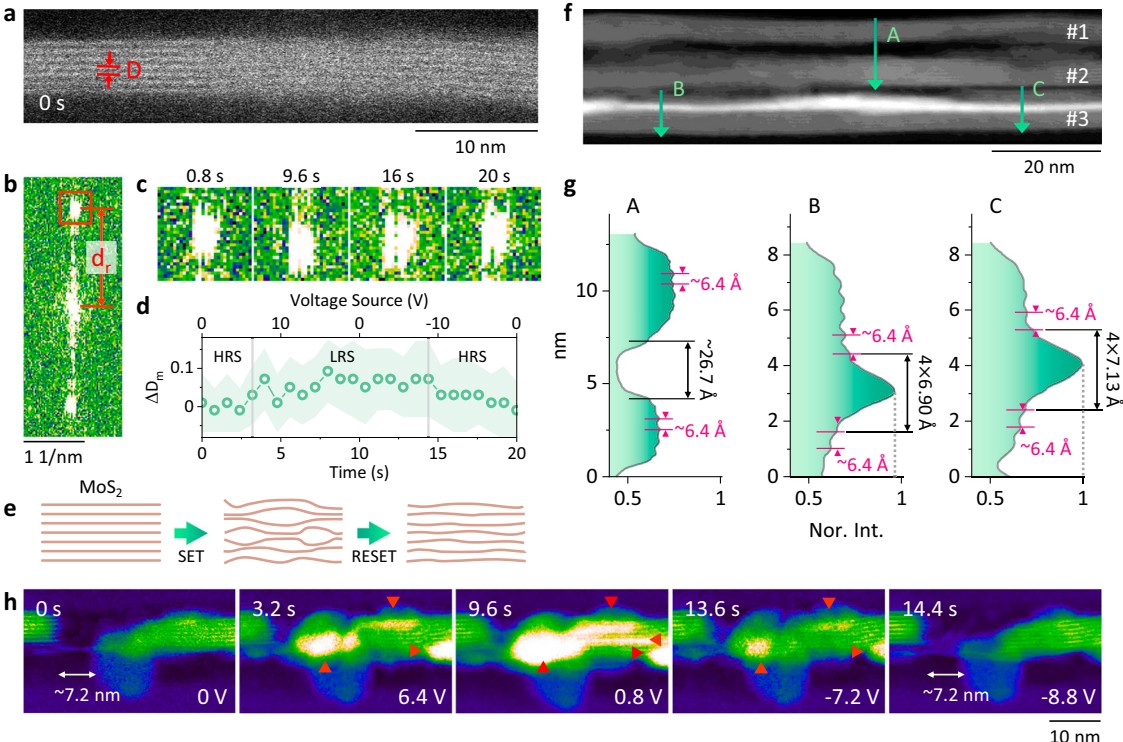

**Fig. 4 | Dynamics of the Ag filaments at the local scale. a** HAADF image from an image series recorded during a 0 V → 10 V → 0 V → −10 V → 0 V bias voltage sweep applied to another lamella (L2). A frame time of 0.8 s was used. The van der Waals (vdW) gap is defined as D. **b** False color fast Fourier transform (FFT) from (**a**). **c** Cropped FFTs corresponding to the region outlined by the square in (**b**) at different times from the image series. **d** The relative changes in $D_m$ (the mean value of $D$ as defined in (**a**) from each HAADF image), $\Delta D_m$, estimated from the image series as a function of time. **e** Proposed scenario of the structural variation of $MoS_2$ during switching. **f** HAADF image from L1 acquired after the 0 V → 5 V → 0 V sweep in Fig. 2c with a frame time of ~25 s. The $MoS_2$ bundles are labeled as #1–3. **g** Intensity profiles along the three arrows labeled as A, B, and C in (**f**). **h** Selected false color HAADF images from an image series recorded during a 0 V → 10 V → 0 V → −10 V → 0 V sweep applied to L2, showing the Ag filament dynamics around locally disrupted $MoS_2$ layers. The corresponding bias voltage are labeled at the bottom. The red triangles mark the emerging contrast from Ag filaments.

outlined by the dashed rectangle in Fig. 5d. Moreover, a miniscule decrease in ΔA2 from 0.088 to 0.087 is also observed at -2.96 V, as shown in the inset in Fig. 5e. Thus, the switching behavior is very sensitive to Ag filament formation, and even small fluctuations of ~0.1% affect the I-V curves noticeably. After this small drop, ΔA2 in Fig. 5e continues increasing again, as does the measured current in Fig. 5d. Another sudden current drop in Fig. 5d is observed at ~−1.1 V. However, no significant change is observed at the corresponding position in Fig. 5e. Thus, a local breakdown of the Ag filaments may account for this phenomenon.

During cycle #3, a sudden jump in current (inset in Fig. 5g) can be well correlated with an abrupt jump in ΔA2 at ~3 V in Fig. 5h. The rapid increase in ΔA2 caused by SET is also evident in the images in Fig. 5m. Within ~0.4 s, ΔA2 increases from ~−0.01 to 0.06, with much brighter contrast, as denoted by the triangles at the bottom of the image. In Fig. 5h, ΔA2 drops sharply at ~32.8 s and −4.6 V after maintaining a high value for ~6.6 s. Two images during the drop are shown in Fig. 5n. The bright contrast, indicated by the triangles, disappears almost completely after RESET, and the value of ΔA2 changes from 0.06 to −0.01 within ~0.4 s.

The I-V curve in Fig. 5j from cycle #4 is similar to that in Fig. 5g, except for a rather bumping RESET. Figure 5k then compares the ΔA2 values between cycles #3 and #4. Unlike the sharp drop in cycle #3 (the transparent curve in Fig. 5k), the ΔA2 from cycle #4 starts dropping much earlier but more slowly, in ~1.8 s to decrease from 0.07 to −0.01. Several images during the RESET from cycle #4 are shown in Fig. 5o. The final value of ΔA2 is consistent for both cycles #3 and #4 (~−0.01), implying that nearly all the Ag in the channel dissolved after RESET.

## Evolution of the Ag electrode

The ΔA4 values in Fig. 5i and l share a similar pattern, whereas the value from cycle #3 is consistently higher than that from cycle #4. Several images around the Ag electrode from cycle #3 are shown in Fig. 5p, with the corresponding time and ΔA4 value noted. The shrinkage and expansion of the electrode are evident. Judging by the image contrast and shape of the electrode, the thinner front end of the electrode is rather active for the switching, as significant contrast increases (i.e., the formation of Ag filaments as indicated by the triangles) are noticed between the Ag electrode front end and $MoS_2$.

As shown by the shadowed peaks in Fig. 5i and l, a slight increase (P1 and P3) before SET and a relatively larger decrease (P2 and P4) after RESET are constantly visible along ΔA4. Moreover, as suggested by the glowing lines, the ΔA4 value is, in fact, stable if the shadows are excluded. The asymmetric peaks (P1 < P2 and P3 < P4) during each cycle lead to a continuous decrease in ΔA4, as observed in Fig. 5i and l. The origin of the peaks can be understood as follows. Before SET and driven by positive bias voltage, a small amount of Ag can migrate inward from the inner part of the Ag electrode (outside A4) to A4, as sketched in the inset in Fig. 5l. Owing to the small bias and relatively stable inner part of Ag, both P1 and P3 are small. Similarly, after RESET and driven by negative bias voltage, the still active Ag within A4 can easily migrate outward, resulting in relatively larger P2 and P4. The net effect would be a decreasing A4 and a growing channel between the two electrodes. Along with the growing channel, higher threshold voltage for SET, larger energy consumption, slower switching, as well as poor stability would be expected for the device. In addition, starting from cycle #2 in Fig. 5, the ΔA2 value at the end of each cycle is negative but relatively stable among the cycles (~−0.01). It is also possible that

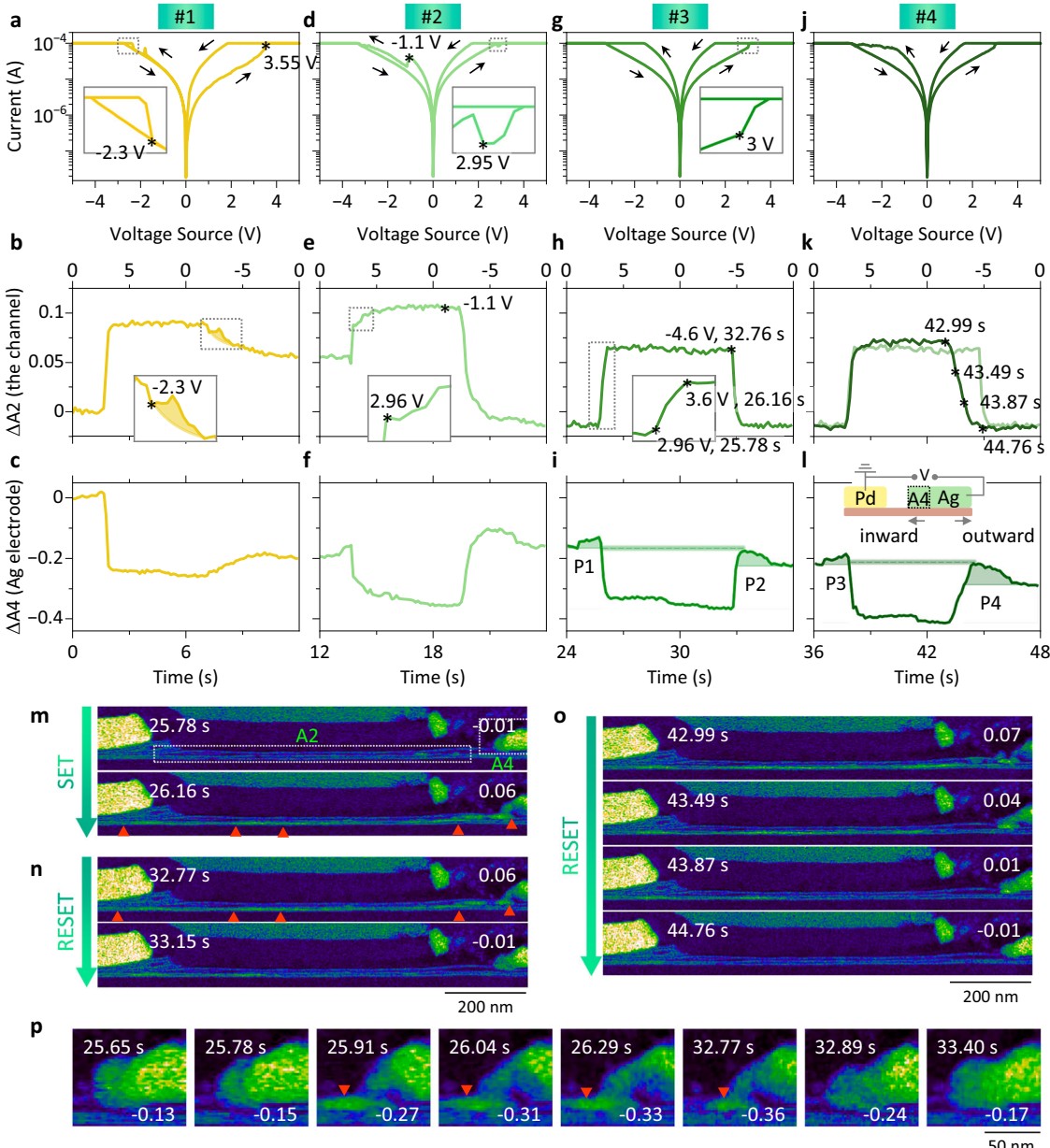

**Fig. 5 | Comparing four consecutive 0 V → 5 V → 0 V → −5 V → 0 V sweeps applied to L1. a–c**, *I-V* curves, and the intensity variations within A2 and A4 (as defined in **m**) during cycle #1. **d–f, g–i, j–l** Corresponding results from cycles #2 to #4. All the insets are corresponding to the regions outlined by the dotted rectangles. An anomalous jump of ΔA2 is highlighted by the shadow in (**b**). The shadowed peaks, P1–P4 in (**i** and **l**) are caused by the rising and dropping ΔA4 before and after the resistive switching (RS), while the glowing lines suggest a constant A4 during the RS. **m, n** False color HAADF images acquired during the SET and RESET processes of cycle #3. The ΔA2 values are noted on the right. **o** False color HAADF images acquired during the RESET process of cycle #4. The ΔA2 values are noted on the right. **p** Enlarged HAADF images showing the evolution of the Ag electrode during cycle #3. The ΔA4 values are noted at the bottom. The red triangles mark the emerging contrast from Ag filaments.

contaminations exist within the lamella before biasing, and contribute to the image intensity. During switching, heat is generated, which could remove the contamination and result in a reduced image intensity (negative ΔA2, Supplementary Note 8 and Supplementary Fig. 12). Once these contaminants are entirely removed, the ΔA2 value at the end of each sweep will no longer be affected. Such an effect could also contribute to the initial decrease in A4, as shown in Fig. 3b. Nevertheless, the heat-induced Ag loss during switching should only be a minor effect in our case (Supplementary Note 9 and Supplementary Fig. 13).

The two lamellas (L1 and L2) investigated in this work have similar channel sizes (~ 0.8 μm wide and <100 nm thick in the y direction as

defined in Fig. 1e). The only difference is the amount of $MoS_2$. L1 in Fig. 1f has three bundles of $MoS_2$, whereas L2 in Fig. 4a only has a single bundle. Each bundle contains typically 5–10 layers of $MoS_2$. Thus, L1 provides not only more, but also different migration pathways for the Ag filaments, i.e., through the space between $MoS_2$ bundles. These spaces are usually several nm wide, which is ideal for the formation of thick Ag filaments, as shown in Fig. 2h, where a ~12 nm thick filament can be identified. Such thick filaments are usually associated with higher switching current, and tend to become permanent after long-term cycling. In contrast, the single $MoS_2$ bundle in L2 restricts the Ag filaments to traverse through the ~6.4 Å wide vdW gaps. Owing to the limited size and thus the limited amount and thickness of Ag filaments,

the Ag dynamics can be interpreted only indirectly by the variation in $D$, as shown in Fig. 4a–g (Supplementary Note 6 and Supplementary Fig. 10). This difference in the quantity and size of the Ag filaments affects the switching behavior as well (Supplementary Fig. 14). L1 is switched with a lower voltage but suffers from a higher current, while the ON/OFF ratio is comparable between the two cases (~one order of magnitude). Thus, by engineering the "distance" between neighboring $MoS_2$ layers[48], fine-tuning of the resistive switching could be expected.

We have visualized the dynamics of Ag filament formation during resistive switching in $MoS_2$-based lateral memristive devices via advanced *operando* TEM. The successful formation of Ag CFs during the SET process and their complete dissolution after the RESET process were explicitly demonstrated by EDXS elemental mapping. Simultaneous HAADF imaging and biasing were applied, where the change in image contrast can be directly associated with Ag CF dynamics during switching. Various sizes of Ag CFs, ranging from a few Å to more than 10 nm, have been observed, and different migration pathways of these CFs have been identified: along the $MoS_2$ surface, through the vdW gaps, across disrupted $MoS_2$ layers, and into adjacent $MoS_2$ bundles. These observations provide unique insights into cycle-to-cycle variabilities, addressing phenomena such as failed switching, slow RESET, and anomalous current fluctuations. The Ag electrode functioned as a reservoir for filament formation and rupture. In addition, internal Ag migration can be driven by small bias voltages before and after resistive switching, leading to a growing channel between electrodes with continuous biasing. Our findings present solid evidence supporting the ECM mechanism underlying resistive switching in $MoS_2$-based memristors. They provide nanoscale insights into CF dynamics that allow the devising of strategies for device optimization, such as tailoring the switching mediums, optimizing the device architecture, and applying proper control during operation. With enhanced resolution, future implementations of our methodology could explore variations in the channel length, tuning the distance of vdW crystallites, incorporating diverse 2D materials and heterostructures, assessing device endurance over multiple cycles, and visualizing filaments from a top-view perspective.

## Methods

Details of the device fabrication can be found in ref. 42. $MoS_2$ was grown on a 2" sapphire substrate via MOCVD in an AIXTRON reactor. The $MoS_2$ film was wet transferred onto $2 \times 2$ cm² Si chips with 275 nm thermal $SiO_2$ using a deionized water process. The metal electrodes were patterned with AZ5214E JP photoresist via optical lithography (EVG 420 Mask Aligner) and laser writing (Microtech LW405C). Pd (50 nm) and Ag (50 nm)/Al (50 nm) electrodes were deposited by electron beam evaporation (FHR Anlagenbau GmbH) followed by lift off in acetone. Finally, the channels were etched via reactive ion etching with a $CF_4/O_2$ chemistry (Oxford Instruments Plasma Lab 100), followed by resist stripping in acetone at 60 °C for 1 h. An additional 80 nm $Al_2O_3$ layer was deposited by electron beam evaporation (FHR Anlagenbau GmbH) to cover the $MoS_2$ channels.

FIB lamellas were prepared from the fabricated memristors via an FEI Helios Nanolab 660 dual-beam microscope with Ga ions and fixed on a Cu FIB lift-out grid. TEM bright field imaging and thickness mapping of the lamellas were performed with an FEI Tecnai F20 instrument at an accelerating voltage of 200 kV. The *operando* experiments were conducted utilizing an STM-TEM holder (Nanofactory Instruments) in an FEI Titan G2 80–200 ChemiSTEM microscope with an accelerating voltage of 200 kV. The microscope is equipped with an extreme-brightness cold field emission gun (XFEG), a probe Cs corrector, a super-X EDXS system, and a Gatan Enfinium ER (model 977) spectrometer with DUAL EELS acquisition capability. A Keithley 2602A source meter was used to apply the bias voltage sweeps during the *operando* measurement. The convergence semi-angle for STEM imaging and EDXS chemical mapping was approximately 22 mrad, whereas the collection semi-angles were 80–200 mrad for HAADF imaging. EDXS maps were typically collected for approximately 30 min, and background subtraction was performed. An iterative rigid alignment algorithm was applied to correct the sample/image drift within each image series, and no image processing was applied.

## Data availability

The Source Data underlying the figures of this study are available with the paper. All raw data generated during the current study are available from the corresponding authors upon request. Source data are provided with this paper.

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

## Acknowledgements

This work has been supported by the Bundesministerium für Bildung und Forschung (BMBF) within NEUROTEC II (16ME0399, 16ME0400, 16ME0398K, and 16ME0403) and NeuroSys (03ZU1106AA and 03ZU1106AD). Z.W. and M.C.L. acknowledge support from the European Union and Chips JU under the Horizon Europe project ENERGIZE (101194458). The authors acknowledge support from Dr. Marcus Hans and Prof. Jochen Schneider (StrucMatLab, RWTH Aachen University) and Dr. Holger Kalisch and Prof. Andrei Vescan (CST, RWTH Aachen University) for providing the MoS$_2$ samples.

## Author contributions

K.R. and M.C.L. conceived this work, designed the experiments, and analyzed the data. S.C. fabricated the devices and selected a suitable chip through separate electrical measurements. K.R. prepared the FIB lamellas. K.R. and J.J. performed the *operando* TEM experiments. Z.W., R.D.B., and J.M. provided insightful discussions on the results. The manuscript was written by K.R. and M.C.L. with contributions from all the authors.

## Funding

## Competing interests

The authors declare no competing interests.
