## [Transparent Peer Review file · Nature Communications]

Unraveling the dynamics of conductive filaments in MoS₂-based memristors by *operando* transmission electron microscopy

Corresponding Author: Professor Max Lemme

Version 0:

Reviewer comments:

Reviewer #1

(Remarks to the Author)

In the manuscript titled “Unraveling the dynamics of conductive filaments in MoS₂-based memristors by *operando* transmission electron microscopy,” Ran et al. present in situ STEM observations of the resistive switching process in a MoS₂ memristive device. The authors have demonstrated a careful in situ TEM experiment, including sample preparation and data acquisition, and they have conducted a comprehensive investigation into how Ag filaments form and dissolve during device operation. While this study is of some interest, many of the reported results remain in the preliminary stage, and certain conclusions are not fully supported by the presented data. Moreover, the overall novelty of this work is questionable unless more atomic-level evidence is provided to solidify the claims. Consequently, I cannot recommend publication until several critical concerns are adequately addressed.

Detailed Comments:

1. On page 9, the authors state: “Overall, the Dm first increased to 10% and then slowly recovered... Thus, the increased Dm can be associated with the Ag filaments formed along the vdW gaps in the LRS.” They also reference image intensity variations to support this conclusion. However, the diameter of an Ag atom is about 300 pm. Even a single-layer Ag chain intercalated into a vdW gap would require a 50% expansion of the interlayer spacing in MoS₂ to enable a continuous Ag filament. Such an expansion should be clearly visible in HRSTEM images (given that the apparent size of Mo atoms is easily distinguishable in atomic-resolution HAADF-STEM images of MoS₂). From Fig. 4 and Supplementary Figure 7, there is no noticeable (>50%) increase in the MoS₂ interlayer distance at any region during switching. Thus, the observed ~10% expansion may not stem from Ag intercalation.
2. Ag is heavier than Mo, meaning that Ag present within a vdW gap in HAADF-STEM imaging should appear brighter than surrounding Mo–S layers. However, in Supplementary Figure 7 and Supplementary Video 2, there is no obvious bright contrast indicating Ag intercalation between layers. A slight contrast increase alone does not convincingly demonstrate Ag infiltration into the vdW gaps.
3. On page 10, the authors indicate that “the bright contrast within bundle #3 suggests the existence of Ag filaments, and peaks with widths of ~27.6 Å (4 × 6.90 Å) and ~28.5 Å (4 × 7.13 Å) are estimated along B and C, respectively. Thus, the intercalation of Ag into adjacent MoS₂ layers can expand the vdW gaps up to 11%.” However, a ~3 nm-wide bright peak cannot be unambiguously attributed to Ag intercalation; the corresponding regions do not display any recognizable layered structure or prominent Mo–S peaks, which might be expected if these regions were merely expanded MoS₂ layers. It is more likely that this peak corresponds to a wider gap between bundles (~3.4 nm from Figure 1i) filled with Ag after switching.
4. The authors also state, “thus, the higher the peak is, the thicker the Ag filament, and ultimately, the larger the expansion of D.” As pointed out above, if a continuous Ag filament had indeed formed inside the vdW gaps, one would expect (i) a clear interlayer expansion visible in HAADF images and (ii) an additional bright contrast peak between two Mo layers, given that Ag is heavier than Mo. So far, none of the provided data convincingly demonstrate this behavior.
5. Supplementary Figure 9 shows that the majority of residual Ag in sample L2 resides at the MoS₂ top surface rather than being located within the MoS₂. If Ag filaments primarily formed by intercalation within the MoS₂ vdW gaps, it is difficult to reconcile why so much Ag would remain on the top surface. Moreover, the authors only show a small portion of the 0.8 μm channel in L2. If Ag is mostly at the top surface, then how does it form filaments within the vdW gaps throughout the channel?

6. Overall, there is insufficient evidence to confirm that the vdW gaps of MoS₂ are hosting the migration of Ag atoms to create conductive filaments. More direct observations, such as higher-resolution imaging and EDS/EELS analyses, are needed to conclusively determine the presence and distribution of Ag within these vdW gaps.
7. On page 13, the authors suggest that “during switching, heat is generated, which could remove the contamination and result in a reduced image intensity.” Since the working power and joule heating of the device can be estimated based on the I–V curve, it would be helpful for the authors to provide a quantitative temperature estimate for the device during operation to support this claim.
8. To gain a deeper understanding of Ag–MoS₂ interactions and Ag migration pathways, atomic-level characterization is needed. Clarifying how the switching process modifies the MoS₂ structure and creates defects at the atomic scale is particularly important. While the manuscript clearly demonstrates that Ag filaments form and dissolve, the broader scientific significance of these findings remains unclear without additional insights into the underlying mechanisms and structural changes.

Reviewer #2

(Remarks to the Author)

In this work, the authors investigate the formation and dissolution of Ag conductive filaments (CFs) in Pd/MoS₂/Ag planar memristors using operando transmission electron microscopy. The study reveals the formation of CFs with a wide range of sizes, from several Ångströms to tens of nanometers. The authors observed different pathways for the formation of Ag filaments, including along the MoS₂ surfaces, within the van der Waals gap between MoS₂ layers, and through the spacing between MoS₂ bundles. This study is interesting and has been conducted with care, shedding light on the key parameters that influence the switching performance of the devices.

- (1) In the main text, the authors demonstrate the distribution of Ag in the memristors through EDS mapping, which confirms the different pathways of CFs. However, is there more evidence to support the composition of CFs being pure Ag? Since the EDS mapping results only show the element distribution and cannot conclusively confirm the specific composition of the filaments. Moreover, Supplementary Note 1 indicates the possible reaction between Ag and S, raising the possibility that the composition of the CFs could be silver sulfide instead.
- (2) What is the direction of the filament growth and dissolution? Furthermore, why is there an Ag signal observed between the MoS₂ bundles at the Ag electrode side in Fig. 1f? Does this mean that Ag has already migrated into the gaps of the MoS₂ bundles before the electrical measurements were conducted?
- (3) The shadowed areas in Fig. 3b. shows a small jump of ~2% and a drop of ~5% before and after switching, respectively. What is the underlying cause of this observed phenomenon? Does this phenomenon have the same reasons as described in the caption of Fig. 5?
- (4) The authors state on page 11, “This suggests that more Ag filaments can form unexpectedly even during the RESET process and cause abrupt current jumps in the I–V curve.” However, the reason for the abrupt current jumps in the I–V curve may not be reasonable or lack direct evidence. During the RESET process, the migration of Ag ions may fill the gaps between the discontinuous filaments, which could also potentially cause an increase in current.
- (5) On page 13, the authors suggest that a reduced image intensity (negative ΔA_2) should result from the removal of contamination during the switching process. However, the authors do not provide details on the composition of this proposed contamination. More evidence would be needed to confirm the existence and nature of this contamination.

Reviewer #3

(Remarks to the Author)

In this manuscript, the authors utilize operando transmission electron microscopy to observe the formation, migration, and dissolution of Ag conductive filaments (CFs) in MoS₂-based memristors. The CFs exhibited a wide range of sizes and followed distinct pathways: along the MoS₂ surface, within the van der Waals gaps between MoS₂ layers, and through the spacing between MoS₂ bundles. Through synchronized HAADF imaging and I–V measurements, the dynamic changes in the Ag CFs and its impact on cycle-to-cycle variability were revealed, elucidating the resistive switching mechanism. My concerns questions on the manuscript are as follows:

1. Does the size of the Ag electrode and the number of MoS₂ layers affect the formation rate of conductive filaments?
2. Does the distance between electrodes matter? How does this influence the filament formation dynamics and switching performance?
3. During long-term cycling, do the paths or morphology of the conductive filaments further change, leading to electrode degradation or permanent filament "shorting"?
4. The Ag electrode acts as a reservoir for CFs during the switching process, with channel D widening observed under continuous biasing. How does this channel D widening affect the long-term stability and durability of the device?
5. The manuscript demonstrates cycle-to-cycle variability in the formation and dissolution of CFs. The author shall provide strategies to reduce this variability to enhance the performance and reliability of the device.

Reviewer comments:

Reviewer #1

(Remarks to the Author)

In their response and revised manuscript, the authors have attempted to address some of the raised questions. However, most major concerns remain unaddressed, and the authors have declined to conduct additional experiments. Furthermore, some of the responses lack internal consistency. Consequently, I cannot recommend publication unless the authors provide solid experimental evidence, particularly regarding atomic-level observations of Ag filament intercalation into the MoS₂ van der Waals gaps.

Detailed Comments:

1. The authors cite literature (Nat Commun 10, 275 (2019)) indicating that Li ion intercalation into bilayer graphene induces a vdW gap expansion of approximately 9%. However, it should be noted that in this reference, Li ions do not form a continuous metallic sheet with atomic stacking akin to Li metal. In contrast, for the Ag filament in this work to be conductive, it must be continuous and maintain a metallic structure—at minimum, a single atomic layer of Ag (as the authors depicted in Supplementary Figure 8e). This scenario is fundamentally different from the intercalation of individual metal atoms into 2D material vdW gaps. For example, the insertion of a silicene monolayer has been reported to increase the interlayer separation between MoS₂ layers from 3 Å to 6.52 Å, corresponding to an expansion of 3.52 Å (Beilstein J. Nanotechnol. 2017, 8, 1952–1960). Similarly, even under ideal assumptions, intercalation of a single atomic layer of Ag metallene would result in a comparable increase in MoS₂ interlayer separation, far exceeding 10%. Therefore, the current experimental observations do not support the claim of Ag conductive filaments intercalation into the MoS₂ vdW gaps.
2. In response to the second question, the authors state that “the MoS₂ is ~100 nm thick, while the formed Ag filaments can be as thin as 2.52 nm.” How do the authors confirm that the Ag filaments can be as thin as 2.52 nm? Have any 3D reconstructions or electron ptychography experiments been performed to determine the three-dimensional distribution of Ag filaments within the MoS₂ vdW gaps?
3. The authors further claim that if comparable amounts of Ag and Mo are present, the Ag contrast should be approximately 1.2 times that of Mo. Let us consider the scenario in Fig. 4g (images b, c), where a 3 nm-wide gap is purportedly comprised of Ag filaments intercalated MoS₂ vdW gaps. Assuming the 11% gap expansion could accommodate a single atomic layer of Ag metallene (which is unlikely), and that the Ag metallene filament completely fills the vdW gap, the amounts of Ag and Mo in a MoS₂/Ag/MoS₂ unit would be roughly comparable. Why, then, does the 3 nm-wide gap in Fig. 4g (images b, c) exhibit a much higher contrast than MoS₂ (possibly 1.5 times greater)? Does this imply the presence of a much thicker Ag filament within the MoS₂ vdW gaps? How could an 11% gap expansion accommodate such a thick Ag filament? Additionally, why is the center of the 3 nm-wide gap significantly brighter than the side regions if the degree of Ag filament intercalation is assumed to be uniform (both 11% gap expansion) within this gap?
4. In response to the sixth question, the authors state that their in situ STM-TEM holder only allows for single-axis tilting, making precise zone-axis alignment for high-resolution imaging impossible. However, this limitation is not a sufficient excuse. The FIB-prepared half-grid sample used in the in situ STM-TEM holder can also be mounted on a conventional double-tilt TEM holder for further characterization. Given that the high-resistance state (HRS) and low-resistance state (LRS) are nonvolatile, it is reasonable to transfer the specimen between holders for atomic-resolution imaging and spectroscopic analysis to determine whether Ag filament intercalation occurs within the MoS₂ vdW gaps, and to study the interaction between Ag and MoS₂ layers after switching. The sample can then be returned to the in situ STM holder for further electrical measurements. Numerous studies have reported high-quality atomic-resolution cross-sectional images of MoS₂, so capturing Ag-intercalated MoS₂ vdW gaps, if present, should not be particularly challenging.
5. The authors argue that MoS₂ is highly sensitive to electron beams and that structural damage can be easily induced, emphasizing that EDXS may destroy the MoS₂ structure. However, I note that the authors used a 200 kV accelerating voltage, which is more damaging to 2D materials than lower voltages. The FEI Titan G2 80-200 TEM is capable of operating at voltages as low as 80 kV, which is commonly used for high-resolution imaging of 2D materials to minimize beam damage without significantly compromising resolution. Why did the authors not attempt high-resolution imaging and EDXS analysis at 80 kV?
6. While the authors have demonstrated in situ switching of planar MoS₂ memristive devices and observed dynamic behaviors of Ag filaments along MoS₂ surface and bundles, the planar device configuration is not commonly used in current neuromorphic systems due to integration challenges in 2D arrays. The research community is more interested in vertical device architectures, which remain unexplored in this study.

Reviewer #2

(Remarks to the Author)

The authors have answered all my comments in a satisfactory way and I am pleased to recommend this manuscript for publication.

Reviewer #3

(Remarks to the Author)

The authors have thoroughly addressed my previous concerns. They provided additional experimental evidence (e.g. EDXS spectra, and comparative HAADF imaging) to clarify Ag filament dynamics and vdW gap expansion, supplemented by theoretical discussions on electrode size and channel length effects. While atomic-scale resolution of Ag intercalation remains challenging due to technical constraints, the revised manuscript now offers a more robust interpretation of the data, supported by new supplementary figures and notes. I recommend acceptance in its current form.

Shaobo Cheng

Version 2:

Reviewer comments:

Reviewer #1

(Remarks to the Author)

The authors have provided detailed responses to my questions and have revised the manuscript accordingly. While some of the explanations are primarily theoretical, I acknowledge the challenges associated with conducting atomic-resolution characterizations. Overall, I believe the paper is suitable for publication at this stage.

Responses to the Reviewer's Comments

We thank you for the swift review process and for dedicating your valuable time to evaluating the manuscript. We also thank you for your constructive and critical comments and suggestions. We have addressed your comments in a point-by-point response below. You can find your comments in black and our response in blue text. In the manuscript, we highlighted the changes in red. The additions and changes in the manuscript are also quoted below for your reference. We hope that our answers suffice to address the points raised and that our manuscript is now ready to be published in Nature Communications.

Max Lemme and Ke Ran (on behalf of all authors)

Reviewer #1 (Remarks to the Author):

In the manuscript titled “Unraveling the dynamics of conductive filaments in MoS₂-based memristors by operando transmission electron microscopy,” Ran et al. present in situ STEM observations of the resistive switching process in a MoS₂ memristive device. The authors have demonstrated a careful in situ TEM experiment, including sample preparation and data acquisition, and they have conducted a comprehensive investigation into how Ag filaments form and dissolve during device operation. While this study is of some interest, many of the reported results remain in the preliminary stage, and certain conclusions are not fully supported by the presented data. Moreover, the overall novelty of this work is questionable unless more atomic-level evidence is provided to solidify the claims. Consequently, I cannot recommend publication until several critical concerns are adequately addressed.

Response: We thank the reviewer for their time and efforts, as well as the positive comments. To the best of our knowledge, this is the first *operando* TEM study on functional 2D material-based memristor. With sufficient spatial and temporal resolution, we successfully captured the conductive filaments (CFs) dynamics during operation and correlated them with device performance for the first time. While atomic-level evidence can surely advance our understanding of the switching process, we would argue that there is a fundamental trade-off between spatial and temporal resolution, which prohibits high spatial resolution in our *operando* measurement. One other limiting factor is the severe sample damages associated with high-resolution techniques for this particular case. Please find our point-by-point responses below.

Detailed Comments:

1. On page 9, the authors state: “Overall, the Dm first increased to 10% and then slowly recovered... Thus, the increased Dm can be associated with the Ag filaments formed along the vdW gaps in the LRS.” They also reference image intensity variations to support this conclusion. However, the diameter of an Ag atom is about 300 pm. Even a single-layer Ag chain intercalated into a vdW gap would require a 50% expansion of the interlayer spacing in MoS₂ to enable a continuous Ag filament. Such an expansion should be clearly visible in HRSTEM images (given that the apparent size of Mo atoms is easily distinguishable in atomic-resolution HAADF-STEM images of MoS₂). From Fig. 4 and Supplementary Figure 7, there is no noticeable (>50%) increase in the MoS₂ interlayer distance at any region during switching. Thus, the observed ~10% expansion may not stem from Ag intercalation.

Response: We thank the reviewer for pointing this out, and agree that the much wider and unoccupied vdW gap of MoS₂ (D, ~640 pm) will allow the intercalation of a single-layer Ag chain (diameter of Ag ion ~252 pm). As a result, D will be expanded.

By analyzing the fast fourier transform (FFT) from a section of multilayer MoS₂, in Fig. 4d, we monitored the evolution of averaged D (D_m) during a successful resistive switching (RS), where an expansion of D_m up to 10% was determined. Besides, the D_m evolved coincidentally with the image intensity, while the image intensity could be directly associated with the formation and dissolution of Ag filaments. Therefore, we attributed the 10% expansion to the Ag intercalation. A similar effect was reported for the Li ion intercalation into bilayer graphene by studying the lithium-storage progress in graphite¹. The diameter of Li ion and the vdW gap of graphene are ~152 pm and ~335 pm, respectively. After the intercalation of a single-layer Li chain, an expansion of the vdW gap was estimated as ~9%, comparable to our measurement.

As suggested by the reviewer, it would be interesting to quantitatively evaluate the intercalation process via numerical calculations. Details, such as the activation energy for the Ag intercalation, the maximum capacity of each vdW gap for Ag ions, the caused reversible/irreversible structural deformation of MoS₂, and possible bonding between Ag and S ions would be critical to advance our understanding of the switching mechanism. Although it is beyond the scope of the current study, they are important directions for our follow-up work.

Accordingly, we have included the discussion above into **Supplementary Note 6**, to comment on the 10% increase of D_m.

2. Ag is heavier than Mo, meaning that Ag present within a vdW gap in HAADF-STEM imaging should appear brighter than surrounding Mo–S layers. However, in Supplementary Figure 7 and Supplementary Video 2, there is no obvious bright contrast indicating Ag intercalation between layers. A slight contrast increase alone does not convincingly demonstrate Ag infiltration into the vdW gaps.

Response: We agree with the reviewer that the Ag (atomic number 47) is heavier than Mo (atomic number 42), and there is no obvious bright contrast in Supplementary Fig. 7 (updated as Supplementary Fig. 8 after revision) or Supplementary Video 2 indicating Ag intercalation between MoS₂ layers.

The HAADF contrast varies with $\sim Z^{1.6}$ (Z is the atomic number). If comparable amounts of Ag and Mo exist in the system, the Ag contrast is ~1.2 times that of the Mo contrast. Along the e-beam direction (y direction as defined in Fig. 1e), the MoS₂ is ~100 nm thick, while the formed Ag filaments can be as thin as 2.52 nm. Such significant difference in the amounts of Mo and Ag could primarily account for the unrecognizable Ag contrast in Supplementary Fig. 8, together with the noisy HAADF images due to short frame time and the possibly insufficient temporal resolution.

Based on the successful RS (Supplementary Fig. 8b) and the coincident evolutions of D_m and image intensities (Supplementary Fig. 8c-d), it is a reasonable interpretation that Ag CFs were formed and traversed along the vdW gaps within the section of multilayer MoS₂ imaged in Supplementary Fig. 8a, causing the expansion of D_m.

Accordingly, we have included this discussion in **Supplementary Note 6** to explain the absent contrast from Ag in Supplementary Fig. 8a.

3. On page 10, the authors indicate that “the bright contrast within bundle #3 suggests the existence of Ag filaments, and peaks with widths of $\sim 27.6 \text{ \AA}$ ($4 \times 6.90 \text{ \AA}$) and $\sim 28.5 \text{ \AA}$ ($4 \times 7.13 \text{ \AA}$) are estimated along B and C, respectively. Thus, the intercalation of Ag into adjacent MoS₂ layers can expand the vdW gaps up to 11%.” However, a $\sim 3 \text{ nm}$ -wide bright peak cannot be unambiguously attributed to Ag intercalation; the corresponding regions do not display any recognizable layered structure or prominent Mo-S peaks, which might be expected if these regions were merely expanded MoS₂ layers. It is more likely that this peak corresponds to a wider gap between bundles ($\sim 3.4 \text{ nm}$ from Figure 1i) filled with Ag after switching.

Response: We agree with the reviewer that based on Fig. 4f alone, the $\sim 3 \text{ nm}$ -wide bright peaks cannot be unambiguously attributed to Ag intercalation, and it is possible that these peaks correspond to wide gaps between MoS₂ bundles which were filled with Ag after switching. Therefore, we provide additional evidence for our claims, both in our response here, in the manuscript, and in Supplementary Note 7 and Supplementary Fig. 9.

The HAADF image in Fig. 4f was recorded from the middle part of L1, where L1 was left in the LRS. The same region was also recorded when L1 was in the HRS before applying any bias voltage, in Fig. 1f. A direct comparison is thus shown in Fig. R1. Fig. R1a is reproduced from Fig. 1f, and its middle region, as outlined by the rectangle, is further enlarged in Fig. R1b. Fig. R1c is a reproduction of Fig. 4f. Clearly, in Fig. R1b, three bundles can be distinguished. The contrast change along each bundle (the parallel direction) is most likely due to the channeling effect caused by the different orientations of each MoS₂ layer within the bundle. As denoted by the read arrows, around the bright peaks in Fig. R1c, there was no gap with $\sim 3 \text{ nm}$ width detected in Fig. R1b before the RS. As also demonstrated in Fig. 2g, the emerging strong image contrast after RS can be directly associated with the formation of Ag CFs. Thus, the $\sim 3 \text{ nm}$ -wide bright peaks in Fig. 4g can be attributed to the Ag intercalation into the vdW gaps after RS. A similar case is shown in Fig. 4h, i.e. in the middle image in Fig. 4h and indicated by the red triangle on the far right. Additionally, as outlined by the red rectangles in Fig. R1b-c, Ag did fill the gap between bundle #2 and #3 after switching as suggested by the reviewer.

It should be also noted, that Fig. R1a was recorded with a much lower spatial resolution comparing with Fig. R1c, to include the whole channel of L1 into the field of view. During our *operando* TEM experiment, we always tried to minimize high-resolution imaging, as MoS₂ is rather sensitive to electron beam, and structural damages can be easily induced. Nevertheless, the comparisons between Fig. R1b and R1c provide solid evidence to exclude the possibility of bundle gaps along arrow B and C before the switching.

Accordingly, we have included this comparison as new **Supplementary Note 7** and **Supplementary Fig. 9**, and revised the main text **Page 10, Paragraph 2**, as follows:

“The bright contrast within bundle #3 suggests significant formation of Ag CFs after SET. These formed Ag CFs may intercalate into the vdW gaps (Supplementary Note 7 and Supplementary Fig. 9) and lie on the surface of the MoS₂ (parallel to the x-z plane as defined in Fig. 1e). The resulted contrast then leads to the strong and broad peaks along B and C, which are measured as $\sim 27.6 \text{ \AA}$ ($4 \times 6.90 \text{ \AA}$) and $\sim 28.5 \text{ \AA}$ ($4 \times 7.13 \text{ \AA}$), respectively. Thus, up to 11% expansion of the vdW gap can be estimated due to the intercalation of Ag into adjacent MoS₂ layers, in line with the estimation in Fig. 4d.”

Figure R1. (a-b) The HAADF image reproduced from Fig. 1f, and an enlarged image from the middle region outlined by the rectangle. (c) A reproduction from Fig. 4f. The arrows and rectangles mark the same locations in (b-c).

4. The authors also state, “thus, the higher the peak is, the thicker the Ag filament, and ultimately, the larger the expansion of D.” As pointed out above, if a continuous Ag filament had indeed formed inside the vdW gaps, one would expect (i) a clear interlayer expansion visible in HAADF images and (ii) an additional bright contrast peak between two Mo layers, given that Ag is heavier than Mo. So far, none of the provided data convincingly demonstrate this behavior.

Response: We thank the reviewer for pointing this out, and agree that the bright peaks along B and C in Fig. 4g are rather broad without individual peaks corresponding to each MoS₂ layers or the Ag filaments between neighboring MoS₂ layers.

As explained in the Response to point 3, these ~3 nm-wide bright peaks can be reasonably attributed to the Ag intercalation into vdW gaps, and an averaged expansion of D up to 11% was estimated. Also noticed in Fig. 4g is that that the bright peaks are much higher than the MoS₂ peaks nearby (marked by the pink lines), suggesting significant Ag CFs formed after SET. The formed Ag CFs may intercalate into the vdW gaps, and lie on the surface of MoS₂ (parallel to the x-z plane as defined in Fig. 1e). The resulted Ag contrast could be then strong enough to suppress the individual MoS₂ peaks, leading to broad peaks and making a direct measurement of the interlayer expansion unavailable.

Accordingly, we have included this discussion into the new **Supplementary Note 7**, revised **Paragraph 2 on Page 10** based on the discussion above as quoted in our response to point 3, and updated **Fig. 4**.

5. Supplementary Figure 9 shows that the majority of residual Ag in sample L2 resides at the MoS₂ top surface rather than being located within the MoS₂. If Ag filaments primarily formed by intercalation within the MoS₂ vdW gaps, it is difficult to reconcile why so much Ag would remain on the top surface. Moreover, the authors only show a small portion of the 0.8 μm

channel in L2. If Ag is mostly at the top surface, then how does it form filaments within the vdW gaps throughout the channel?

Response: We thank the reviewer for pointing this out. Supplementary Fig. 9b (updated as Supplementary Fig. 11b after revision) does indeed show significant Ag residuals at the top surface of MoS₂, as well as around the disrupted MoS₂ on the right. Such significant Ag residuals on the surface can be reasonably associated with the disrupted MoS₂ on the right, where the migrating Ag CFs can easily “go out of” the MoS₂ during SET, but somehow failed in going back during RESET and resided there.

The observation in Supplementary Fig. 11b was presented to demonstrate the possibility of residual Ag after a full switching cycle, which could account for a failed RESET (in Supplementary Fig. 11a). However, it doesn't exclude the possibility that Ag filaments can migrate and/or reside within the vdW gaps. Experimentally, it is challenging to map Ag within the vdW gaps, due to the minimum amount of Ag which could be possibly accommodated by the narrow vdW gaps, and the severe (and prohibiting) sample damages caused by long EDXS elemental mapping.

Regarding the migrations of Ag CFs, diverse pathways have been observed: along the MoS₂ surfaces, within the vdW gap between MoS₂ layers, and through the spacing between MoS₂ bundles. We have no evidence showing that the vdW gap is the primary option for all the cases. To avoid misleading the readers, we revised the sentence on Page 9 Line 5, “The hypothesis is that Ag filaments traverse mainly through the vdW gaps,...” to “The hypothesis for this local region is that Ag filaments traverse mainly through the vdW gaps,...”.

As also pointed out by the reviewer, Supplementary Fig. 11 only shows a small portion of the channel in L2 after one RS. For a full view of the channel in L2 during another successful switching cycle, we would refer to Supplementary Fig. 13. For this specific case, a limited number of Ag filaments was expected to migrate mainly along the vdW gaps, as no significant contrast increase was observed on the surface of MoS₂.

6. Overall, there is insufficient evidence to confirm that the vdW gaps of MoS₂ are hosting the migration of Ag atoms to create conductive filaments. More direct observations, such as higher-resolution imaging and EDS/EELS analyses, are needed to conclusively determine the presence and distribution of Ag within these vdW gaps.

Response: We agree with the reviewer that it is important to have high-resolution images and spectra to confirm the Ag migration through the vdW gap of MoS₂.

High-resolution imaging with atomic columns resolved often needs a certain frame time (several seconds) for high image quality, and the sample must be aligned along one of its low order zone axes. For the memristors in our study, the RS usually takes 10 ~ 20 s. Thus, the relatively long frame time for high-resolution imaging is in conflict with the required temporal resolution for capturing the CFs dynamics. Besides, during the device fabrication, the MoS₂ was transferred to the Si/SiO₂ substrate with random orientation, while the in situ TEM holder is a single-tilt one (only α -tilt is allowed). Thus, it is extremely challenging, in fact, mostly impossible to orient the MoS₂ to the desired zone axis for high-resolution imaging. Besides, in order to resolve the Ag filaments within individual vdW gaps, the amount of formed Ag filaments must be appropriate. Too much Ag will lead to the broad peaks as in Fig. 4g, while too little Ag

might be below the detection limit. Unfortunately, the amount of Ag CFs formed during RS can not be so accurately controlled yet.

Therefore, we carefully selected the imaging parameters, aiming to extract most information out of our experiment with as high as possible resolution. For example, 0.8 s/frame, 46.21pm/pixel, 1024 pixels were used for Fig. 4a. The resulted images allowed sufficient spatial resolution to resolve the individual MoS₂ layers and thus to estimate the D expansion, as well as sufficient temporal resolution to track the evolution of D as a function of time during RS. Based on the successful RS, the coincident evolutions of image intensity (mostly likely caused by the emerging Ag CFs) and D, and the reasonable expansion of D, we concluded that vdW gaps can host the Ag migration during RS.

The typical EDXS and EELS elemental mapping take more than 20 mins, in order to collect sufficient signals. Along with this long exposure time, one critical issue is the sample damages caused by electron beam, especially for the beam sensitive 2D materials under a high-resolution condition. Fig. R2 compares a local region from L2 before and after EDXS elemental mapping, showing significant structural degradation of the MoS₂ already in 3 mins. Thus, we only applied EDXS elemental mapping with a relatively low resolution as shown in Fig. 2d-f. These maps proved explicitly the formation and dissolution of Ag CFs without causing significant sample damages. EELS usually requires even longer dwell time at each pixel, which causes sample damages even higher than that from EDXS. Moreover, in our case, the EELS edges of Ag, Mo and S are either broad, delayed or located at a rather high energy loss region of >2000 eV. Thus, it is rather challenging to detect these edges accurately, making EELS unlikely an ideal technique for this particular case.

Due to the challenges mentioned above, we minimized the application of high-resolution techniques in this study, and found a good compromise among spatial resolution, temporal resolution, and sample damages to support our findings. Meanwhile, we do realize that unique insights down to atomic scale will largely advance our understanding of the RS process. In our ongoing study, we are working on different approaches to improve both spatial and temporal resolution, while keep the sample damages within a limited range. However, this is beyond the scope of the current study, which aimed to demonstrate the RS in 2D-material-based memristors via *operando* TEM for the first time, to the best of our knowledge.

Accordingly, the discussion on high-resolution techniques has been included in **Supplementary Note 6**, and the conclusion part on **Page 15** has been revised.

Figure R2. Comparison of the same region before and after EDXS elemental mapping of ~3 mins.

7. On page 13, the authors suggest that “during switching, heat is generated, which could remove the contamination and result in a reduced image intensity.” Since the working power and joule heating of the device can be estimated based on the I–V curve, it would be helpful for the authors to provide a quantitative temperature estimate for the device during operation to support this claim.

Response: We thank the reviewer for pointing this out. Below, we estimate the device temperature under operation using a very approximate physical model.

Taking L1 for example, the length and width of the MoS₂ conductive channel are $L \approx 800$ nm and $W \approx 100$ nm, and the thickness of MoS₂ is $t \approx 20$ nm (Supplementary Note 2 and Supplementary Fig. 2). As the mass density of multi-layer MoS₂ is about $\rho = 5.06 \times 10^6$ g/m³,² a total mass of MoS₂ of $m = \rho \cdot L \cdot W \cdot t = 8.1 \times 10^{-15}$ g is obtained. Besides, the specific heat capacity (per unit cell) at room temperature (300 K) of MoS₂ is $C_m = 0.3$ J·g⁻¹·K⁻¹, which can be expressed by

$$C_m = (Q_j - Q_c) / (m \cdot \Delta T), \quad \text{Eq. 1}$$

where Q_j is the Joule heat from the switch cycle (1.75×10^{-3} J based on the I - V curve $\int I \cdot V \cdot dt$, taking the curve in Fig. 2b for example), and Q_c is the heat conducted away through MoS₂. For simplicity, the highest temperature was assumed at the center of the MoS₂, and the generated heat was dissipating from there to both electrodes which were at room temperature (300 K). ΔT is the temperature difference between the center of the MoS₂ channel and the electrodes.

The in-plane thermal conductivity of MoS₂ is $\kappa = 44$ W·m⁻¹·K⁻¹.³ Assuming the time span is roughly the same as the cycle time (~ 10 sec), κ can be expressed by

$$\kappa \cdot 10 \text{ sec} = Q_c \cdot (L/2) / (W \cdot t \cdot \Delta T) \quad \text{Eq. 2}$$

Solving Eq.1 and Eq.2 together, $\Delta T = 795$ K is estimated, resulting in ~ 1065 K at the center of the MoS₂ channel, which is in line with the reports in literature^{4,5}.

However, it should be noted that this rough calculation is based on significant simplifications of the actual system. The parameters of MoS₂ (such as C_m and κ) are actually temperature-dependent, and the estimated ΔT is only an averaged value over one bias sweep. In order to accurately assess the evolution of device temperature during RS, dedicated numerical calculation using finite element method is necessary.

Accordingly, we have included this discussion as the new **Supplementary Note 8**, and revised the main text on **Page 13, Paragraph 2** to refer to the added Supplementary Note 8.

8. To gain a deeper understanding of Ag-MoS₂ interactions and Ag migration pathways, atomic-level characterization is needed. Clarifying how the switching process modifies the MoS₂ structure and creates defects at the atomic scale is particularly important. While the manuscript clearly demonstrates that Ag filaments form and dissolve, the broader scientific significance of these findings remains unclear without additional insights into the underlying mechanisms and structural changes.

Response: We agree with the reviewer that understanding the CFs dynamics and their interactions with MoS₂ will largely benefit from high-resolution information down to atomic

scale. To the best of our knowledge, our work is the first *operando* TEM study on functional 2D material-based memristors, which resolved the CFs during operation. With sufficient spatial and temporal resolution and well-managed sample damages, we successfully captured the CFs dynamics and correlated them with the device performance for the first time. Based on this success, we are currently working on different approaches to enhance the imaging resolution, which will be further combined with numerical calculations to advance our understanding of the CFs dynamics and the underlying mechanisms.

Reviewer #2 (Remarks to the Author):

In this work, the authors investigate the formation and dissolution of Ag conductive filaments (CFs) in Pd/MoS₂/Ag planar memristors using *operando* transmission electron microscopy. The study reveals the formation of CFs with a wide range of sizes, from several Ångströms to tens of nanometers. The authors observed different pathways for the formation of Ag filaments, including along the MoS₂ surfaces, within the van der Waals gap between MoS₂ layers, and through the spacing between MoS₂ bundles. This study is interesting and has been conducted with care, shedding light on the key parameters that influence the switching performance of the devices.

Response: We thank the reviewer for their time and efforts, as well as the positive comments. Please find our point-by-point responses and corresponding revisions below.

(1) In the main text, the authors demonstrate the distribution of Ag in the memristors through EDS mapping, which confirms the different pathways of CFs. However, is there more evidence to support the composition of CFs being pure Ag? Since the EDS mapping results only show the element distribution and cannot conclusively confirm the specific composition of the filaments. Moreover, Supplementary Note 1 indicates the possible reaction between Ag and S, raising the possibility that the composition of the CFs could be silver sulfide instead.

Response: We thank the reviewer for pointing this out. As described in Supplementary Note 1, after exposure to the ambient air and light, we observed structural degradation of the lamella, where the EDXS elemental mapping results suggested the possible formation of silver sulfide. To avoid such degradation, all the lamellas used in our study were freshly prepared and constantly protected from ambient air and light before the *operando* TEM. As shown by the HAADF image and EDXS composite map in Fig. 2d (a fresh lamella for *operando* TEM), the device structure is well preserved.

To further confirm the composition, an EDX spectrum from the filament is extracted and plotted in Fig. R3 (the top row). Fig. R3a is reproduced from Fig. 2f, and the dashed rectangles in Fig. R3c defined the region for the spectrum in Fig. R3e. In addition to the Cu K peak (from the TEM grid), a strong Ag L peak together with a small S K peak show up (see the inset for details), suggesting that the filaments are mainly composed of Ag. To understand the origin of the small S K peak, EDX spectrum from the same region before the RS is also plotted in Fig. R3 (the bottom row). Fig. R3b is reproduced from Fig. 2d, and the dashed rectangles in Fig. R3d defined the region for the spectrum in Fig. R3f. There, in spite of the empty region, a small S K peak shows up again. Moreover, the heights of both S K peaks in Fig. R3e and R3f are

comparable. Thus, the small S peak in Fig. R3e can be safely attributed to the surrounding MoS₂ (most likely due to delocalization of the electron beam during EDXS elemental mapping), and the filaments detected in our case are pure Ag.

Accordingly, this discussion is included as the new **Supplementary Note 4** and **Supplementary Fig. 5**, and the text in the main text **Page 7 Line 2** "...confirms significant Ag signals between the MoS₂ bundles." has been revised to "...confirms significant Ag signals, thus Ag filaments, between the MoS₂ bundles (Supplementary Note 4 and Supplementary Fig. 5).".

Figure R3. (a-b) Reproductions from Fig. 2f and 2d. (c-d) Enlarged HAADF image and composite map from the regions outlined by the rectangles in (a-b). (e) EDX spectrum extracted from the region outlined by the dashed rectangle in (c). Inset shows details around the Ag peak. (f) EDX spectrum extracted from the region outlined by the dashed rectangle in (d).

(2) What is the direction of the filament growth and dissolution? Furthermore, why is there an Ag signal observed between the MoS₂ bundles at the Ag electrode side in Fig. 1f? Does this mean that Ag has already migrated into the gaps of the MoS₂ bundles before the electrical measurements were conducted?

Response: For memristors based on electrochemical metallization (ECM) mechanism, the CFs form through the dissolution and re-deposition of the active electrode materials, as illustrated in Fig. R4a⁶. The mechanism is typically described to have the following steps: (A) oxidation and dissolution of the active electrode (Ag/Cu) under positive bias; (B) migration of Ag/Cu ions; (C) reduction at the inert electrode; (D-E) nucleation and filamentary growth; and (F) the final transition from a high resistance state (HRS) to a low resistive state (LRS). Following this theory, the Ag CFs in our case should grow from the Pd to the Ag electrode, and dissolve starting from the weak filaments.

As Fig. 1f is a HAADF image, we assume that the reviewer was asking about the sparse Ag intensities as indicated by the arrow in Fig. R4b, which is a reproduction of Fig. 1g. These

sparse Ag signals together with the Ag particle (outlined by the circle in Fig. R4b) disappeared after a successful RS (Supplementary Fig. 3b), as shown in Fig. R4c (reproduced from Fig. 2d). Therefore, we would expect some slight Ag redeposition close to the Ag electrode from the FIB preparation. These redeposited Ag loosely lay on the surface of the lamella, and can be easily removed by the heat and/or mechanical instability generated during the RS, similar to the outlined Ag particle. Therefore, it is unlikely that Ag already migrated into the MoS₂ bundles before the electrical measurements.

Accordingly, we have included this discussion into the new **Supplementary Note 6 and 8** and **Supplementary Fig. 7 and 12**, and revised the main text **Page 13, Paragraph 2** to refer to the added Supplementary Note 8.

Figure R4. (a) The ECM mechanism⁶. (b-c) Reproductions from Fig. 1g and Fig. 2d. One bias voltage sweep was applied between the two maps.

(3) The shadowed areas in Fig. 3b. shows a small jump of ~2% and a drop of ~5% before and after switching, respectively. What is the underlying cause of this observed phenomenon? Does this phenomenon have the same reasons as described in the caption of Fig. 5?

Response: A small jump before the switching and a small drop after the switching are consistently observed for ΔA_4 , as in Fig. 3b, 5f, 5i, and 5l. These phenomena can be reasonably explained by Ag migration driven by small bias voltage before SET and after RESET as described in the caption of Fig. 5.

(4) The authors state on page 11, “This suggests that more Ag filaments can form unexpectedly even during the RESET process and cause abrupt current jumps in the I-V curve.” However, the reason for the abrupt current jumps in the I-V curve may not be reasonable or lack direct evidence. During the RESET process, the migration of Ag ions may fill the gaps between the discontinuous filaments, which could also potentially cause an increase in current.

Response: We agree with the reviewer that during the RESET process, the migration of Ag ions may fill the gaps between the discontinuous filaments, and cause an increase in current.

As defined by the dotted rectangle in Fig. 5m, A₂ corresponds to the intensity integrated within the whole channel. Thus, bridging discontinuous filaments by already existed Ag ions will not cause an increase in A₂. Instead, such an increase as indicated by the shadowed region in Fig. 5b suggested the formation of extra Ag within the channel, which coincided with the current

jump at ~ 2.3 V as shown in the inset in Fig. 5a. Therefore, we correlated the current jump and the A2 increase.

Accordingly, we have revised the sentence on Page 12 Line 1 from "...can form unexpectedly even during the RESET..." to "...can form unexpectedly (since A2 corresponds to the intensity integrated within the whole channel) even during the RESET...".

(5) On page 13, the authors suggest that a reduced image intensity (negative $\Delta A2$) should result from the removal of contamination during the switching process. However, the authors do not provide details on the composition of this proposed contamination. More evidence would be needed to confirm the existence and nature of this contamination.

Response: We thanks the reviewer for pointing it out. We would expect two possible sources for the contamination: redeposition from FIB and hydrocarbon contamination.

As demonstrated in the Response to point 2, redeposited Ag loosely lying on the surface of the lamella can be removed during initial bias voltage sweeps, which leads to a reduced image intensity. Besides, hydrocarbon contaminations are rather common for TEM samples, and plasma clean is normally applied to get rid of them. In our case, no plasma clean was used, due to the oxygen involved during the cleaning which could potential react with the Ag electrode. Thus, a certain hydrocarbon contamination was expected at the sample surface. During STEM imaging, the residual hydrocarbon contaminations will normally accumulate to the area irradiated by the electron beam and cause extra bright contrast. However, we didn't observe such feature even during imaging with high resolution. Thus, most of the hydrocarbon contaminations should be already removed during the initial bias voltage sweeps, contributing to the reduced image intensity as well.

Accordingly, we have included this discussion as the new **Supplementary Note 8** and **Supplementary Fig. 12**, and revised the main text **Page 13, Paragraph 2** to refer to the added Supplementary Note 8.

Reviewer #3 (Remarks to the Author):

In this manuscript, the authors utilize operando transmission electron microscopy to observe the formation, migration, and dissolution of Ag conductive filaments (CFs) in MoS₂-based memristors. The CFs exhibited a wide range of sizes and followed distinct pathways: along the MoS₂ surface, within the van der Waals gaps between MoS₂ layers, and through the spacing between MoS₂ bundles. Through synchronized HAADF imaging and I–V measurements, the dynamic changes in the Ag CFs and its impact on cycle-to-cycle variability were revealed, elucidating the resistive switching mechanism. My concerns questions on the manuscript are as follows:

Response: We thank the reviewer for their time and efforts, as well as the positive comments. Please find our point-by-point responses and corresponding revisions below.

1. Does the size of the Ag electrode and the number of MoS₂ layers affect the formation rate of conductive filaments?

Response: We thank the reviewer for pointing this out. For the two lamellas in our study, we did not notice any size effect of the Ag electrode, as we intentionally kept the sizes of both lamellas similar to enable a direct comparison between them.

Theoretically, for lateral memristors based on 2D materials and following the ECM mechanism, the size of electrode could affect the formation rate of CFs critically⁷. Smaller electrodes should enhance the formation rate resulting from elevated current densities, constrained filament growth, and localized heating. However, excessive heat will also lead to material degradation, while too small electrodes might cause insufficient ion supply. Larger electrodes can provide a more stable ion supply and better heat dissipation, which are beneficial for the stability of the devices. In contrast, a small ion supply (reservoir) may prevent the growth of thick (strong) filaments and enhance endurance⁸.

As discussed above, the rate of CFs formation depends on the electric field, temperature, and the availability of the ions, while the vdW gaps of MoS₂ offer migration paths for these CFs. Thus, a larger number of MoS₂ layers could facilitate the CFs migration, which is potentially beneficial to accelerate the CFs formation. Overall, the electrode size should be optimized together with the 2D materials' properties and the device geometry (particularly the channel size for lateral device) for achieving the desired performance. However, these are device-related considerations that go beyond the scope of this *operando* TEM-focused paper, but which we explore in other works, e.g.⁹⁻¹¹.

2. Does the distance between electrodes matter? How does this influence the filament formation dynamics and switching performance?

Response: We thank the reviewer for pointing this out. For the two lamellas in our study, we intentionally kept the channel lengths similar to enable a direct comparison between them. However, based on our previous study, we consider the channel length as a critical factor affecting the filament formation dynamics and switching performance.

In principle, devices with long channel lengths require higher “on-threshold” voltages ($V_{t,on}$), including rather high “forming” voltages for the first switching cycle (see e.g.⁹ or our recent preprint¹²). They also show slower switching speeds, and have increased energy consumption. In contrast, short channels enable faster and more localized formation with reduced variability, but may suffer from over-forming, and the resulting device failure. Overall, scaling down channel lengths is favorable for energy efficiency and switching speed, while the balance between low-voltage operation and device endurance should be taken into consideration. Based on our electrical measurements⁹, lateral devices with channel lengths < 2.1 μm showed RS without a forming step, while devices with channel lengths > 3.9 μm required larger initial voltages between 10 ~ 50 V, and $V_{t,on}$ between 2 ~ 3 V was reported for all the devices. In the literature¹³⁻¹⁷, a forming process is usually required for similar 2D material-based lateral memristors with various channel lengths (between 10 nm and 15 μm), and different $V_{t,on}$ ranging from ~0.4 V up to 20 V were reported.

Thus, for lateral devices, both filaments formation and device performance are rather sensitive to the channel lengths. As described in the conclusion part, varying the channel lengths is one important direction for our following *operando* TEM studies.

Accordingly, on **Page 13 Paragraph 2**, we have added the following sentence: “Along with the growing channel, higher threshold voltage for SET, larger energy consumption, slower switching, as well as poor stability would be expected for the device.”

3. During long-term cycling, do the paths or morphology of the conductive filaments further change, leading to electrode degradation or permanent filament "shorting"?

Response: We thank the reviewer for pointing this out. Certain changes in the paths or morphology of the CFs after long-term cycling are usually expected, which is one of the challenges in minimizing cycle-to-cycle variability of memristors. In fact, a major theoretical advantage of 2D materials as the switching medium for memristors is the more confined formation of CFs ideally along the vdW gap, comparing with the traditionally utilized 3D transition metal oxides, although the topic still requires substantial research. To further tailor the 2D material quality, research efforts are continuously made, via phase engineering^{18,19}, heterostructures fabrication^{20,21}, and so on. The degradation of the active electrode is generally associated with chemical reactions, over-heating, and the formation of permanent CFs after long-term cycling, leading to deteriorated device performance or even failed devices (not switchable anymore).

Accordingly, revisions were made on **Page 14 Paragraph 2** as follows: “..., where a ~12 nm thick filament can be identified. Such thick filaments are usually associated with higher switching current, and tend to become permanent after long-term cycling. In contrast, ...”.

4. The Ag electrode acts as a reservoir for CFs during the switching process, with channel D widening observed under continuous biasing. How does this channel D widening affect the long-term stability and durability of the device?

Response: We thank the reviewer for pointing this out. In Fig. 4a-e, ~0.1% increase of the vdW gap (D) was observed at the end of one full switching cycle. Such marginal increase was attributed to a small amount of residual Ag ions in the vdW gaps or a slow recovery of the MoS₂ structure after RESET. This channel D widening also suggests the possibility of permanent CFs and/or irreversible structural modifications of MoS₂ after long-term cycling, which could lead to cycle-to-cycle variability, deteriorated device performance, and eventually failed devices.

Accordingly, the sentence on **Page 9 Paragraph 2** “...or a slow recovery of the MoS₂ structure.” has been revised as “...or a slow recovery of the MoS₂ structure, both of which will contribute to deteriorated device performance.”.

5. The manuscript demonstrates cycle-to-cycle variability in the formation and dissolution of CFs. The author shall provide strategies to reduce this variability to enhance the performance and reliability of the device.

Response: We thank the reviewer for pointing it out. For the 2D-material based lateral memristors following the ECM mechanism, to reduce the cycle-to-cycle variability and ultimately to enhance the device performance, strategies from various aspects could be considered:

- a. the optimization of switching mediums (thickness of the 2D materials and the distance between individual layers, heterostructures of different 2D materials, and 2D materials with tailored properties via phase engineering)
- b. the design of device architecture (the shape of electrodes to localize electrical field and to promote filament nucleation, and the length of the conduction channel)
- c. proper control during operation, e.g. current compliance and operation temperature
- d. thermal management (integrating heat-dissipating materials to avoid over-heating during operation)

Accordingly, we have revised the description on **Page 15** “They provide nanoscale insights into CF dynamics that allow the devising of strategies for device optimization.” to “They provide nanoscale insights into CF dynamics that allow devising strategies for device optimization, such as tailoring the switching mediums, optimizing the device architecture, and applying proper control during operation.”.

References:

1. Ji, K. *et al.* Lithium intercalation into bilayer graphene. *Nat. Commun.* **10**, 275 (2019).
2. Thomas, N. *et al.* 2D MoS₂: structure, mechanisms, and photocatalytic applications. *Mater. Today Sustain.* **13**, 100073 (2021).
3. Bae, J. J. *et al.* Thickness-dependent in-plane thermal conductivity of suspended MoS₂ grown by chemical vapor deposition. *Nanoscale* **9**, 2541–2547 (2017).
4. Teja Nibhanupudi, S. S. *et al.* Ultra-fast switching memristors based on two-dimensional materials. *Nat. Commun.* **15**, 2334 (2024).
5. Yang, S. J. *et al.* Volatile and Nonvolatile Resistive Switching Coexistence in Conductive Point Hexagonal Boron Nitride Monolayer. *ACS Nano* **18**, 3313–3322 (2024).
6. Menzel, S., Tappertzhofen, S., Waser, R. & Valov, I. Switching kinetics of electrochemical metallization memory cells. *Phys. Chem. Chem. Phys.* **15**, 6945–6952 (2013).
7. Lv, H. *et al.* Evolution of conductive filament and its impact on reliability issues in oxide-electrolyte based resistive random access memory. *Sci. Rep.* **5**, 7764 (2015).
8. Li, T., Yu, H., Chen, S. H. Y., Zhou, Y. & Han, S.-T. The strategies of filament control for improving the resistive switching performance. *J. Mater. Chem. C* **8**, 16295–16317 (2020).
9. Cruces, S. *et al.* Volatile MoS₂ Memristors with Lateral Silver Ion Migration for Artificial Neuron Applications. *Small Sci.* **n/a**, 2400523.
10. Völkel, L. *et al.* Influence of humidity on the resistive switching of hexagonal boron nitride-based memristors. *Npj 2D Mater. Appl.* **9**, 1–6 (2025).
11. Völkel, L. *et al.* Resistive Switching and Current Conduction Mechanisms in Hexagonal Boron Nitride Threshold Memristors with Nickel Electrodes. *Adv. Funct. Mater.* **34**, 2300428 (2024).
12. Cruces, S. *et al.* Volatile and Nonvolatile Resistive Switching in Lateral 2D Molybdenum Disulfide-Based Memristive Devices. Preprint at <https://doi.org/10.48550/arXiv.2504.07979> (2025).
13. Sangwan, V. K. *et al.* Multi-terminal memtransistors from polycrystalline monolayer molybdenum disulfide. *Nature* **554**, 500–504 (2018).
14. Sangwan, V. K. *et al.* Gate-tunable memristive phenomena mediated by grain boundaries in single-layer MoS₂. *Nat. Nanotechnol.* **10**, 403–406 (2015).

15. Farronato, M. *et al.* Memtransistor Devices Based on MoS₂ Multilayers with Volatile Switching due to Ag Cation Migration. *Adv. Electron. Mater.* **8**, 2101161 (2022).
16. Yin, S. *et al.* Emulation of Learning and Memory Behaviors by Memristor Based on Ag Migration on 2D MoS₂ Surface. *Phys. Status Solidi A* **216**, 1900104 (2019).
17. Ding, G. *et al.* Reconfigurable 2D WSe₂-Based Memtransistor for Mimicking Homosynaptic and Heterosynaptic Plasticity. *Small* **17**, 2103175 (2021).
18. Li, Y. *et al.* Anomalous resistive switching in memristors based on two-dimensional palladium diselenide using heterophase grain boundaries. *Nat. Electron.* **4**, 348–356 (2021).
19. Liu, X. *et al.* On-device phase engineering. *Nat. Mater.* **23**, 1363–1369 (2024).
20. Geim, A. K. & Grigorieva, I. V. Van der Waals heterostructures. *Nature* **499**, 419–425 (2013).
21. Novoselov, K. S., Mishchenko, A., Carvalho, A. & Castro Neto, A. H. 2D materials and van der Waals heterostructures. *Science* **353**, aac9439 (2016).

Responses to the Reviewer's Comments

We thank you for the swift review process and for dedicating your valuable time to evaluating the revised manuscript. We also appreciate your constructive and critical comments and suggestions. We have addressed your comments in a point-by-point response below. You can find your comments in black and our response in blue text. In the manuscript, we highlighted the changes in red. We hope that our answers suffice to address the points raised and that our manuscript is now ready to be published in Nature Communications.

Max Lemme and Ke Ran (on behalf of all authors)

Reviewer #1 (Remarks to the Author):

In their response and revised manuscript, the authors have attempted to address some of the raised questions. However, most major concerns remain unaddressed, and the authors have declined to conduct additional experiments. Furthermore, some of the responses lack internal consistency. Consequently, I cannot recommend publication unless the authors provide solid experimental evidence, particularly regarding atomic-level observations of Ag filament intercalation into the MoS₂ van der Waals gaps.

Response: We thank the reviewer for their time and efforts. We fully agree that high-resolution TEM is a powerful tool, from which we have benefited enormously in addressing many different material systems with picometer precision.^{1,2} However, we would like to clearly state that the primary objective of our *operando* study is to demonstrate the real-time observation of the Ag conductive filament (CFs) formation and its dynamics during resistive switching (RS).

Therefore, we had to find a reasonable compromise between temporal resolution, spatial resolution, and sample damage. High-resolution imaging alone cannot fully describe the dynamic process of RS, but inevitably sacrifices temporal resolution and significantly damages the sample. Additionally, the specific requirements of *operando* experiments, such as the special sample geometry, a dedicated *in situ* holder, and the strict requirement on sample freshness, prohibit high-resolution imaging either during or post the *operando* experiment. Please find our point-by-point responses below.

Detailed Comments:

1. The authors cite literature (Nat Commun 10, 275 (2019)) indicating that Li ion intercalation into bilayer graphene induces a vdW gap expansion of approximately 9%. However, it should be noted that in this reference, Li ions do not form a continuous metallic sheet with atomic stacking akin to Li metal. In contrast, for the Ag filament in this work to be conductive, it must be continuous and maintain a metallic structure—at minimum, a single atomic layer of Ag (as the authors depicted in Supplementary Figure 8e). This scenario is fundamentally different from the intercalation of individual metal atoms into 2D material vdW gaps. For example, the insertion of a silicene monolayer has been reported to increase the interlayer separation between MoS₂ layers from 3 Å to 6.52 Å, corresponding to an expansion of 3.52 Å (Beilstein J. Nanotechnol. 2017, 8, 1952–1960). Similarly, even under ideal assumptions, intercalation of a single atomic layer of Ag metallene would result in a comparable increase in MoS₂ interlayer separation, far exceeding 10%. Therefore, the current experimental observations do not support the claim of Ag conductive filaments intercalation into the MoS₂ vdW gaps.

Response: We agree that individual ion intercalation is different from the formation of a conductive film (or an initial step towards CF formation). Nevertheless, we'd like to iterate our previous response, that, based on the successful RS and the coincident evolution of D_m and image intensity, it is reasonable to correlate the expanded vdW gap with Ag intercalation. Fig. 4d is thus able to provide direct and reliable insights into the structural changes of MoS₂ during RS at the nm scale.

Furthermore, we thank the reviewer for sharing the literature.³ In that report, it was claimed that buckled Si clusters (possibly oxidized as evidenced by XPS) were intercalated into bilayer MoS₂. Experimentally, a hill-and-valley surface of MoS₂ with local height differences of ~ 2.5 Å was determined by scanning tunneling microscopy (STM) (Fig. 1f in the literature). Using density functional theory (DFT) calculations, by fixing the lattice constant of MoS₂, an increase of interlayer distance of free-standing bilayer MoS₂ was estimated to be on the order of 2.5-3.2 Å, resulting from intercalated Si clusters (Fig. 7 in the literature). However, we would like to point out that these numbers represent local maxima of the changes in the interlayer distance of free-standing bilayer MoS₂. In our case, the D_m as plotted in Fig. 4d is an averaged value from a section of multilayer MoS₂ (7-layer, ~ 42 nm long, and ~ 100 nm thick), which is sandwiched between Al₂O₃ and SiO₂ as shown in Fig. 1f.

We agree that a significant expansion of D (like the oxidized Si cluster in bilayer MoS₂³) is possible during RS. However, it should be only a local feature in our case, considering the confinement of MoS₂ by Al₂O₃ and SiO₂. To estimate ΔD_m , all these local expansions were averaged within the field of view in Fig. 4a, and also along the direction of the electron beam for ~ 100 nm (i.e., into the y -plane of the figure R1a, defined by the thickness of the lamella). For example, as illustrated in Fig. R1a, during a successful RS, Ag chains may only form in one of the six vdW gaps in Fig. 4a. Thus, a significant local ΔD will be divided by 6 to estimate the averaged ΔD_m . Besides, as suggested in Fig. R1b, within the xy plane, the formed Ag chains may be randomly distributed or broader (limited by the lamella thickness of ~ 100 nm in our *operando* experiments). While the electrons are travelling along the y direction, the imaged ΔD is also an averaged value over the ~ 100 nm along the y direction (~ 40 times of the diameter of the Ag ions of 2.52 Å). Moreover, the MoS₂ structure exhibits a certain degree of flexibility. As proposed in Fig. 4e, the expansion of one vdW gap could lead to the shrinking of the adjacent vdW gaps. Therefore, ΔD_m may be well below its local maximum, and the experimentally measured value of $\sim 10\%$ should be reasonable and can be attributed to the Ag intercalation.

Figure R1. (a) Illustration of Ag chain only formed within one vdW gap of the multilayer MoS₂. (b) Possible distribution of the Ag chain within the xy plane. Electron beam/viewing direction is along the y direction.

As also mentioned in our previous response, in order to quantitatively evaluate the intercalation process, numerical calculations are necessary. Details, such as the activation energy for the Ag intercalation, the maximum capacity of each vdW gap for Ag ions, the caused reversible/irreversible structural deformation of MoS₂, and possible bonding between Ag and S ions are all critical information to advance our understanding of the switching mechanism. Although it is beyond the scope of the current study, they are important directions for our future work. Based on the discussion above, **Supplementary Note 6** and **Supplementary Fig. 8** have been revised accordingly.

2. In response to the second question, the authors state that “the MoS₂ is ~100 nm thick, while the formed Ag filaments can be as thin as 2.52 nm.” How do the authors confirm that the Ag filaments can be as thin as 2.52 nm? Have any 3D reconstructions or electron ptychography experiments been performed to determine the three-dimensional distribution of Ag filaments within the MoS₂ vdW gaps?

Response: We apologize for this critical typo. Our response should have read “as thin as 2.52 Å”. As suggested by the reviewer in comment #1, at least a single atomic layer of Ag should form to account for the observed RS. The diameter of an Ag ion is ~2.52 Å, and the thickness of the lamella is ~100 nm (along the electron beam direction). Thus, the formed Ag filament within vdW gap could be as thin as ~2.52 Å or as thick as ~100 nm along the viewing direction.

3. The authors further claim that if comparable amounts of Ag and Mo are present, the Ag contrast should be approximately 1.2 times that of Mo. Let us consider the scenario in Fig. 4g (images b, c), where a 3 nm-wide gap is purportedly comprised of Ag filaments intercalated MoS₂ vdW gaps. Assuming the 11% gap expansion could accommodate a single atomic layer of Ag metallene (which is unlikely), and that the Ag metallene filament completely fills the vdW gap, the amounts of Ag and Mo in a MoS₂/Ag/MoS₂ unit would be roughly comparable. Why, then, does the 3 nm-wide gap in Fig. 4g (images b, c) exhibit a much higher contrast than MoS₂ (possibly 1.5 times greater)? Does this imply the presence of a much thicker Ag filament within the MoS₂ vdW gaps? How could an 11% gap expansion accommodate such a thick Ag filament? Additionally, why is the center of the 3 nm-wide gap significantly brighter than the side regions if the degree of Ag filament intercalation is assumed to be uniform (both 11% gap expansion) within this gap?

Response: Thank you for elaborating on this point. In Fig. 4g, we observed strong peaks that were ~3 nm wide along arrows B and C, and measured an ~11% expansion of the vdW gap of MoS₂. We interpret this question as follows: why is such a strong contrast from Ag with a peak shape observed in Fig. 4g, while the image intensity is rather uniform in Fig. 4a, where a comparable expansion of D of ~10% was estimated?

As illustrated in Fig. R2, this can be explained by the excess Ag going out of the MoS₂ through its open edges and landing on the surface of the FIB lamella. During RS, the formed Ag filaments can travel freely within the xy plane. When excess Ag filaments are formed, i.e. more than what can be accommodated by the vdW gaps of the MoS₂, one option for them is to go outside of the MoS₂ through its open edges, and accumulate on the surface, see Fig. R1c.

For HAADF imaging, the electron beam is coming from the y direction, and the resulting image contrast is proportional to the amount of material along the y direction. Therefore, both the Ag filaments within the vdW gap and the excess Ag landing on the surface would contribute to the ~3 nm wide strong peaks in Fig. 4g. For the case of Fig. 4a, we would expect much fewer Ag CFs formed during RS (the *I-V* curves in Supplementary Fig. 8b and in Fig. 2c), and thus much less excess Ag landing on the surface.

In the manuscript, on Page 10 Paragraph 2, we briefly described this feature, “These formed Ag CFs may intercalate into the vdW gaps (Supplementary Note 7 and Supplementary Fig. 9) and lie on the surface of the MoS₂ (parallel to the x-z plane as defined in Fig. 1e).” We thank the reviewer for pointing this out. Accordingly, the explanation above has been included in **Supplementary Note 7** and **Supplementary Fig. 9**.

Figure R2. (a) Reproduced from Fig. 4f. (b) Cross-sectional view of the device at LRS, with Ag filaments within the vdW gaps of MoS₂, as well as excess Ag on the top surface. (c) Plan-view of the device illustrated in (b). Here, the Ag filaments are filling the vdW gaps uniformly for simplicity, and the electron beam is coming along y direction.

4. In response to the sixth question, the authors state that their in situ STM-TEM holder only allows for single-axis tilting, making precise zone-axis alignment for high-resolution imaging impossible. However, this limitation is not a sufficient excuse. The FIB-prepared half-grid sample used in the in situ STM-TEM holder can also be mounted on a conventional double-tilt TEM holder for further characterization. Given that the high-resistance state (HRS) and low-resistance state (LRS) are nonvolatile, it is reasonable to transfer the specimen between holders for atomic-resolution imaging and spectroscopic analysis to determine whether Ag filament intercalation occurs within the MoS₂ vdW gaps, and to study the interaction between Ag and MoS₂ layers after switching. The sample can then be returned to the in situ STM holder for further electrical measurements. Numerous studies have reported high-quality atomic-resolution cross-sectional images of MoS₂, so capturing Ag-intercalated MoS₂ vdW gaps, if present, should not be particularly challenging.

Response: As shown in Fig. 1c, the half TEM grid is firmly glued with a Ag wire (on the right side of the TEM grid) in order to be mounted to the STM-TEM holder for *operando* experiments. The gluing alone takes 5 hours (heating may accelerate the process, but was not applied due to the active Ag) to ensure the mechanical stability during the *operando* experiment. For the use of a conventional double-tilt holder, the firmly glued Ag wire must be mechanically detached from the half-grid, a rather risky process, as the half-grid and thus the lamella can be easily deformed. Additionally, the multiple transfers of lamella between different holders need repeated gluing and detaching, which is extremely risky with nearly zero success rate. It also involves constant exposure of the lamella to ambient conditions (see Supplementary Note 1 and Fig. 1 for the strict requirement on the sample freshness). Therefore, the experimental procedure suggested by the reviewer is not feasible for our *operando* study.

As also pointed out by the reviewer, imaging MoS₂ cross-sections with high resolution has been extensively reported. Fig. R3 shows one of our HAADF images made for checking the as-grown MoS₂ on sapphire substrate. Here, the orientation of MoS₂ constantly varies, while the sapphire is tilted along its [100] direction. Moreover, as explained in our last response, in order to resolve the Ag filaments within individual vdW gaps, the amount of formed Ag filaments must be appropriate. Too much Ag will lead to the broad peaks as in Fig. 4g, while too little Ag might be undetectable. Unfortunately, the amount of Ag CFs formed during RS cannot be so accurately controlled yet.

Overall, directly imaging the Ag filaments within individual vdW gaps would be highly desirable. However, the top priority of this work was to resolve the CF dynamics with sufficient spatial and temporal resolution in a real memristor during operation. By careful experimental design and data analyses, we are able to present the first *operando* TEM on RS. Based on this success, further improving the imaging resolution will be an interesting direction for future work.

Figure R3. HAADF image of as-grown MoS₂ on sapphire substrate.

5. The authors argue that MoS₂ is highly sensitive to electron beams and that structural damage can be easily induced, emphasizing that EDXS may destroy the MoS₂ structure. However, I note that the authors used a 200 kV accelerating voltage, which is more damaging to 2D materials than lower voltages. The FEI Titan G2 80-200 TEM is capable of operating at voltages as low as 80 kV, which is commonly used for high-resolution imaging of 2D materials

to minimize beam damage without significantly compromising resolution. Why did the authors not attempt high-resolution imaging and EDXS analysis at 80 kV?

Response: As mentioned by the reviewer, numerous dedicated high-resolution TEM studies on MoS₂ have been conducted using 80 kV to minimize the beam damage. To the best of our knowledge, these works are limited to imaging, while high-resolution EDXS mapping of MoS₂, with atomic columns resolved, is rarely reported.

In our *operando* study, beam damage is not the only factor that needs to be considered. To achieve sufficient temporal resolution, a rather short exposure time for imaging was used, which resulted in limited beam damage but largely degraded image quality. Thus, we chose 200 kV instead of 80 kV (200 kV offers better spatial resolution than 80 kV) to reasonably balance the spatial resolution, temporal resolution, and beam damage.

Regarding EDXS elemental mapping with high resolution, beam damages are almost inevitable at both 200 kV and 80 kV, due to the intense scanning over a small region (~several nm) for a long time (>20 mins). Thus, we only applied EDXS elemental mapping with a relatively low resolution as shown in Fig. 2d-f. These maps proved explicitly the formation and dissolution of Ag CFs without causing significant sample damage.

6. While the authors have demonstrated in situ switching of planar MoS₂ memristive devices and observed dynamic behaviors of Ag filaments along MoS₂ surface and bundles, the planar device configuration is not commonly used in current neuromorphic systems due to integration challenges in 2D arrays. The research community is more interested in vertical device architectures, which remain unexplored in this study.

Response: We thank the reviewer for this suggestion. We agree that vertical memristors in cross-bar arrays are being investigated intensely, including our own related work. Vertical devices often switch only at random spots (at grain boundaries or defects) and there, finding a filament in a cross-sectional TEM required tremendous luck and effort. However, here, we used the lateral devices as examples, which are perfectly suitable for carrying out the *operando* TEM study that reveals the Ag CF dynamics in an operating memristive device. This successful demonstration will undoubtedly be extended to the vertical devices in our future work and possibly require a further refined method. We are also certain other groups will follow this example and look forward to those results.

Reviewer #2 (Remarks to the Author):

The authors have answered all my comments in a satisfactory way and I am pleased to recommend this manuscript for publication.

Response: we thank the reviewer for the positive decision.

Reviewer #3 (Remarks to the Author):

The authors have thoroughly addressed my previous concerns. They provided additional

experimental evidence (e.g. EDXS spectra, and comparative HAADF imaging) to clarify Ag filament dynamics and vdW gap expansion, supplemented by theoretical discussions on electrode size and channel length effects. While atomic-scale resolution of Ag intercalation remains challenging due to technical constraints, the revised manuscript now offers a more robust interpretation of the data, supported by new supplementary figures and notes. I recommend acceptance in its current form.

Response: we thank the reviewer for the positive decision.

Reference

1. Ran, K. *et al.* in situ observation of reversible phase transitions in Gd-doped ceria driven by electron beam irradiation. *Nat. Commun.* **15**, 8156 (2024).
2. Ran, K. *et al.* Direct Visualization of Distorted Twin Boundaries in Ce-Doped GdFeO₃. *Nano Lett.* **23**, 2945–2951 (2023).
3. Bremen, R. van *et al.* Intercalation of Si between MoS₂ layers. *Beilstein J. Nanotechnol.* **8**, 1952–1960 (2017).